# Nano-DMS-MaP allows isoform-specific RNA structure determination

Patrick Bohn [1,3], Anne-Sophie Gribling-Burrer[1,3], Uddhav B. Ambi[1] & Redmond P. Smyth [1,2] ✉

Genome-wide measurements of RNA structure can be obtained using reagents that react with unpaired bases, leading to adducts that can be identified by mutational profiling on next-generation sequencing machines. One drawback of these experiments is that short sequencing reads can rarely be mapped to specific transcript isoforms. Consequently, information is acquired as a population average in regions that are shared between transcripts, thus blurring the underlying structural landscape. Here, we present nanopore dimethylsulfate mutational profiling (Nano-DMS-MaP)—a method that exploits long-read sequencing to provide isoform-resolved structural information of highly similar RNA molecules. We demonstrate the value of Nano-DMS-MaP by resolving the complex structural landscape of human immunodeficiency virus-1 transcripts in infected cells. We show that unspliced and spliced transcripts have distinct structures at the packaging site within the common 5′ untranslated region, likely explaining why spliced viral RNAs are excluded from viral particles. Thus, Nano-DMS-MaP is a straightforward method to resolve biologically important transcript-specific RNA structures that were previously hidden in short-read ensemble analyses.

RNA structure is a main determinant of RNA function[1,2], and is controlled largely through the folding of RNA into regions of single-stranded and double-stranded RNA[3]. Among the methods for interrogating RNA folding, chemical probing stands out for its ease of use and ability to determine RNA structure in situ[4,5]. During chemical probing, RNA is treated with reagents that react preferentially with single-stranded regions of RNA. One such reagent, dimethylsulfate (DMS), methylates the N3 position of cytosines and the N1 position of adenines at the Watson–Crick face of unpaired residues, giving rise to information that can be used to perform high-accuracy RNA structure predictions[6–10]. This small cell permeable chemical is used widely for the in situ or in vitro structural analysis of RNA or RNA–protein complexes[11–14]. In classical experiments, the modified nucleotides, 1-methyladenosine (m1A) and 3-methylcytosine (m3C), stall reverse transcription causing reverse transcriptase (RT) drop off to form truncated complementary

DNAs (cDNAs) that can be analyzed by gel or capillary electrophoresis[5]. In DMS sequencing (DMS-seq), truncated cDNAs are subjected to next-generation sequencing to perform genome-wide measurements of RNA structure[15,16]. Alternatively, DMS mutational profiling sequencing (DMS-MaP) uses modified buffer conditions to perform error-prone reverse transcription of DMS-modified nucleotides[12,17,18]. DMS-MaP therefore allows for straightforward measurements of RNA structure by counting mutations.

DMS-MaP can perform high-throughput measurements of RNA structure[12,17] but also has its drawbacks. Most importantly, typical RT conditions produce short cDNA molecules ideal for sequencing on Illumina sequencing machines[19,20]. The resulting reads, however, rarely span whole transcript isoforms like those generated by alternative transcription start and termination sites, or by alternative splicing. Therefore, DMS-MaP is not well suited to identify structural differences

[1]Helmholtz Institute for RNA-based Infection Research, Helmholtz Centre for Infection Research, Würzburg, Germany. [2]Julius-Maximilians-Universität Würzburg, Faculty of Medicine, Würzburg, Germany. [3]These authors contributed equally: Patrick Bohn, Anne-Sophie Gribling-Burrer. ✉e-mail: redmond.smyth@helmholtz-hiri.de

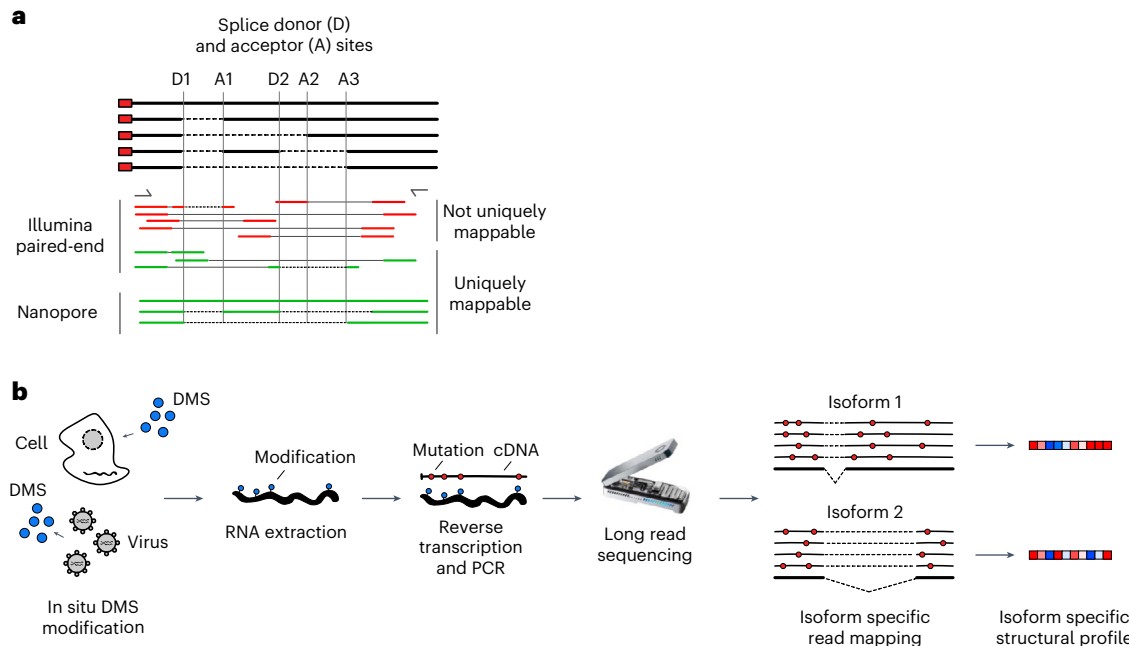

**Fig. 1 | Transcript isoforms structural analysis by Nano-DMS-MaP.** Nano-DMS-MaP for isoform-resolved RNA structure determination. **a**, Transcript isoforms can be generated by alternative splicing. Common regions between transcript isoforms may have different structures, but short sequencing reads cannot always be unambiguously mapped to transcript isoforms. Long-read nanopore sequencing can be uniquely mapped to transcript isoforms. **b**, DMS can probe RNA structure in situ in cells or virions. Modified RNA is extracted and reverse transcribed into long cDNA molecules for sequencing on nanopore devices. Isoform-specific read mapping allows isoform-specific structural profiling. MinION device image credit: Oxford Nanopore Technologies (2023).

in transcript isoforms (Fig. 1a)[21,22]. In humans, transcript isoforms are very common. Over 50% of genes show variability in transcription start site, 70% of genes exhibit alternative polyadenylation and around 95% of multi-exonic genes are alternatively spliced[23]. Consequently, much of the structural information of cellular RNAs obtained by current MaP techniques reflects a population average of distinct underlying structures and isoforms, likely concealing important gene regulatory mechanisms.

Here, we have overcome problems related to the ambiguous mapping of short sequencing reads to transcript isoforms in DMS-MaP experiments by developing nanopore DMS-MaP (Nano-DMS-MaP) (Fig. 1b). We used an ultraprocessive RT enzyme to generate long cDNA molecules with mutational signatures at sites of DMS modification. We also developed an analytical workflow that enables the structural determination of individual transcript isoforms from error-prone nanopore sequencing data. We apply Nano-DMS-MaP to resolve the complex structural landscape of human immunodeficiency virus (HIV)-1 transcripts in infected cells. We show that the genomic and spliced transcripts have distinct structures at their common 5′ untranslated region (UTR) structures, which includes the packaging motif. These structural differences likely explain the exclusion of spliced transcripts from the virion. Thus, we suggest that, in addition to increasing protein diversity, alternative splicing results in the generation of RNA transcripts with distinct functions mediated by altered RNA structures. Our data provide a powerful demonstration that critical regulatory mechanisms can be hidden in short-read ensemble analyses, and that these can be uncovered by long-read RNA structural analysis.

## Results

### Optimization of nanopore long-read sequencing

Sequencing accuracy is critical for DMS-MaP experiments because this method relies on the reverse transcription of DMS adducts into mutations that must be distinguished from site-specific errors by normalization with an unmodified control sample[17,24]. Signal-to-noise ratio

is therefore limited by the inherent error rate of the sequencing platform and by RT and polymerase chain reaction (PCR) errors. DMS-MaP experiments are commonly designed to induce mutation frequencies of 1–2% at A and C residues. Such mutational signatures can be detected by short-read Illumina sequencing where most nucleotides have a Phred quality score (Q-score) above 30 (Q30), which is equivalent to 99.9% accuracy. While nanopore sequencing devices can perform long-read sequencing, they have a magnitude higher error rate than Illumina sequencing machines, with a substantial proportion of reads exhibiting low accuracies[25]. Recent improvements to Nanopore sequencing chemistry and basecalling algorithms, however, have raised modal accuracy to Q20 (99% accuracy)[26].

We first assessed whether we could obtain long cDNA molecules from RNA molecules treated with DMS. To do this, we reverse transcribed and amplified a 532 nucleotide (nt) portion of the unspliced (US) HIV-1 RNA from infected cells. This region comprises the highly structured 5′ UTR and the beginning of the viral *gag* gene, and folds into a series of stem-loop structures that regulate the HIV-1 life cycle[27]. For reverse transcription, we used MarathonRT because it was reported to generate long cDNAs in the presence of RNA modifications[28,29]. Still, we found an inverse relationship between DMS concentration and the amount and length of cDNAs recovered (Supplementary Fig. 1a). We tried to improve cDNA recovery by adjusting parameters such as reverse transcription time, temperature and $Mn^{2+}$ concentrations (Supplementary Fig. 1b,c), but the only parameter that had a substantial effect was DMS concentration. This indicates a trade-off between DMS concentration and maximal recoverable transcript length. Consequently, the amplification of long transcript isoforms is only possible with DMS concentrations below those typically used in DMS-MaP experiments.

We next tested whether nanopore sequencing has a sufficient accuracy to enable high quality structure determination, especially at lower DMS concentrations that are expected to have reduced signal-to-noise ratio. We performed a nanopore sequencing run using Kit 12 chemistry

and evaluated platform- and DMS-induced mutation rates on the US HIV-1 RNA (Methods). The sequencing run generated data with a mean Q-score of 16 (97% accuracy), indicating that global error rates were substantially above the signal in most DMS-MaP experiments. After implementing a median per read Q-score filter of ten to remove the lowest quality sequences, mean mutation rates were 2.5% without DMS, which increased progressively from 2.7% at 8 mM DMS to 3.6% at 85 mM DMS (Supplementary Table 1 and Extended Data Fig. 1a). Accordingly, signal-to-noise ratios ranged from 1.13 at the lowest DMS concentration to 1.75 at the highest concentration (Supplementary Table 2).

Next, we evaluated the quality of structural information by calculating normalized DMS reactivities for each DMS concentration (Methods). We then quantitatively compared the results against a reference structure previously obtained by chemical probing of HIV RNA extracted from virions (Extended Data Fig. 1b)[14]. For this comparison, we used the receiver operator characteristic area under the curve (ROC-AUC) score—a summary statistic to evaluate the correlation of DMS reactivity with strandedness (for example, whether the nucleotide was in single- or double-stranded RNA). A score of 0.5 signifies a random association of the two variables, whereas 1 indicates a perfect match. At the 8 mM DMS concentration, ROC-AUC scores reached 0.6, indicating the presence of low-quality structural information. Increasing the concentration of DMS improved the ROC-AUC scores, reaching 0.9 at 85 mM DMS concentration, indicating excellent agreement with the reference structure. By subsampling reads, we observed that ROC-AUC scores saturated at approximately 4,000 reads, indicating that the low signal could not be overcome by increasing read depths (Extended Data Fig. 1c).

To further improve the recovery of structural information, we systematically optimized filtering parameters. We evaluated an option to ignore insertions and deletions (indels) when counting mutations, as well as median per read Q-score filters and per position Q-score filters (Supplementary Table 2 and Extended Data Fig. 2). Ignoring indels decreased error rates by fivefold in the untreated sample, from 2.5% to 0.5%. The mutation rate in the DMS-treated samples also decreased (3.6% to 1.7% at 85 mM), but the much lower mutation rate in the control led to a substantial increase in signal-to-noise ratio (from 1.12 to 1.61 at 8 mM DMS and from 1.75 to 5.7 at 85 mM DMS) (Supplementary Table 2). Accordingly, ignoring indels improved ROC-AUC scores under almost all conditions, and especially at lower DMS concentrations, which are required to reach the longest read lengths (Extended Data Fig. 3). This observation is explained by a mutation type analysis, which revealed that a high proportion of nanopore sequencing errors are indels (Extended Data Fig. 4a), while DMS-induced mutations were nearly exclusively single nucleotide mismatches (Extended Data Fig. 4b,c). Whereas read median filters greater than Q-score 10 decreased coverage without improving signal-to-noise ratio, the inclusion of a per position filter to remove nucleotide positions with a Q-score of less than 22 led to another notable increase in signal-to-noise ratio (from 1.6 to 3.25 at 8 mM and from 5.7 to 19 at 85 mM) (Supplementary Table 2). Altogether, we identified optimal parameters for Nano-DMS-MaP, namely the discarding of indels, a median per read Q-score filter of 10 (to remove low-quality reads), and a per position Q-score filter of 22. These straightforward data treatment steps gave a three- to tenfold boost in signal-to-noise ratio over raw nanopore data, which translates to higher quality structural information at lower DMS concentrations and coverages (Extended Data Fig. 3 and Supplementary Table 2).

## Nano-DMS-MaP recovers known structures

We next performed structural analysis of the US HIV-1 5′ UTR. Using optimal Nano-DMS-MaP parameters, the global mutation rate was 0.09% in the untreated control, 0.2% at 8 mM DMS and 1.05% at 85 mM DMS (Extended Data Fig. 4d). As expected from the chemical selectivity of DMS, mutations were located principally at A and C residues (Fig. 2a). Calculated DMS reactivities were consistent across all DMS concentrations for the US HIV-1 RNA (Extended Data Fig. 5a) and, when

plotted onto the reference structure, there was a clear correspondence with strandedness (Fig. 2b,c). A reactivity threshold of approximately 0.5 gave the best separation between true and false classifications for the RNA in question (Fig. 2d). ROC-AUC scores of 0.92 indicated near-perfect agreement between our data and the reference structure (Fig. 2e). By subsampling reads, we found that ROC-AUC scores converged towards 0.9 for all DMS concentrations at read depths of 30,000, but similar scores could also be achieved at 4,000 reads for higher DMS concentrations (Fig. 2f). We also identified a highly consistent relationship between the Pearson's correlation of the DMS reactivities of two replicates and their agreement with secondary structure by ROC-AUC, which provides a generally applicable quality control measure for the accuracy of Nano-DMS-MaP data (Fig. 2g and Extended Data Fig. 5b). When comparing the optimized Nano-DMS-MaP analysis against Illumina sequencing of the same cDNA, we observed equivalency in mutation rates and near-perfect agreement of the measured DMS reactivities at equal coverages (Fig. 2h and Extended Data Fig. 5c–f). DMS-guided folding recovered the reference structure at all DMS concentrations, demonstrating that Nano-DMS-MaP can be used for RNA structure determination (Supplementary Data Files 2 and Extended Data Fig. 5f). Mutation type analysis surprisingly revealed slightly higher single nucleotide substitution rates in the Illumina dataset compared with our nanopore data, reinforcing the notion that nanopore errors are mainly indels (Extended Data Fig. 4b,c). This analysis also confirms that MarathonRT nearly exclusively generates single nucleotide substitutions at positions of DMS modification, which enables our simple data filtering steps to boost signal-to-noise ratio without introducing bias (Extended Data Fig. 4).

Next, we tested the general applicability of our workflow on a panel of compact, functionally diverse and highly structured RNAs in vitro (Extended Data Fig. 6). We selected these RNAs because they have complex, yet well-characterized, secondary structures. Furthermore, the three-dimensional structure of several of these RNAs was recently solved using an integrated approach combining information from chemical probing and cryo-electron microscopy experiments[30]. In all cases, Nano-DMS-MaP recovered structural information with ROC-AUC scores of between 0.81 and 0.96 (Fig. 2i). We also performed Nano-DMS-MaP on a well-characterized RNA in situ, selecting the 18S human ribosomal RNA because of its relatively long length (1.9 kilobases (kb)). Again, we obtained useful structural information with a ROC-AUC score of 0.76 at A and C residues, which is a value consistent with other chemical probing studies of ribosomal RNAs (Fig. 2i and Extended Data Fig. 7)[31].

## The structure of the HIV-1 genome in cells and virions

We next assessed the capabilities of Nano-DMS-MaP for long-read structural analysis by in situ probing of the 8.5 kb HIV-1 genome in both infected cells and virions. Although nanopore sequencing itself does not have a theoretical limit on read length, Nano-DMS-MaP includes RT and PCR enzymatic reactions as potential length-limiting steps. To avoid complications during PCR due to the repeat regions used to form the HIV-1 long terminal repeats (LTRs), we performed amplification of the genome in two PCR reactions, each spanning 4 kb (Fig. 3a). Notably, only a single cDNA reaction spanning the whole 8.5 kb of the US RNA was required for both PCR reactions to be successful. Both amplicons could be generated at DMS concentrations of up to 20 mM from less that 0.5 ml of viral supernatant, which demonstrates the sensitivity of Nano-DMS-MaP for long-read RNA structural analysis. DMS reactivities were highly correlated between independent experimental replicates (Pearson's r 0.86–0.98) and DMS concentrations (Pearson's r 0.92 in cell, 0.98 in virion) (Extended Data Fig. 8a,b). The slightly lower correlation the between replicates obtained in the cell despite their similar coverages may indicate structural flexibility and/or the presence of alternative structures in the cell that are not present in virions (Extended Data Fig. 8).

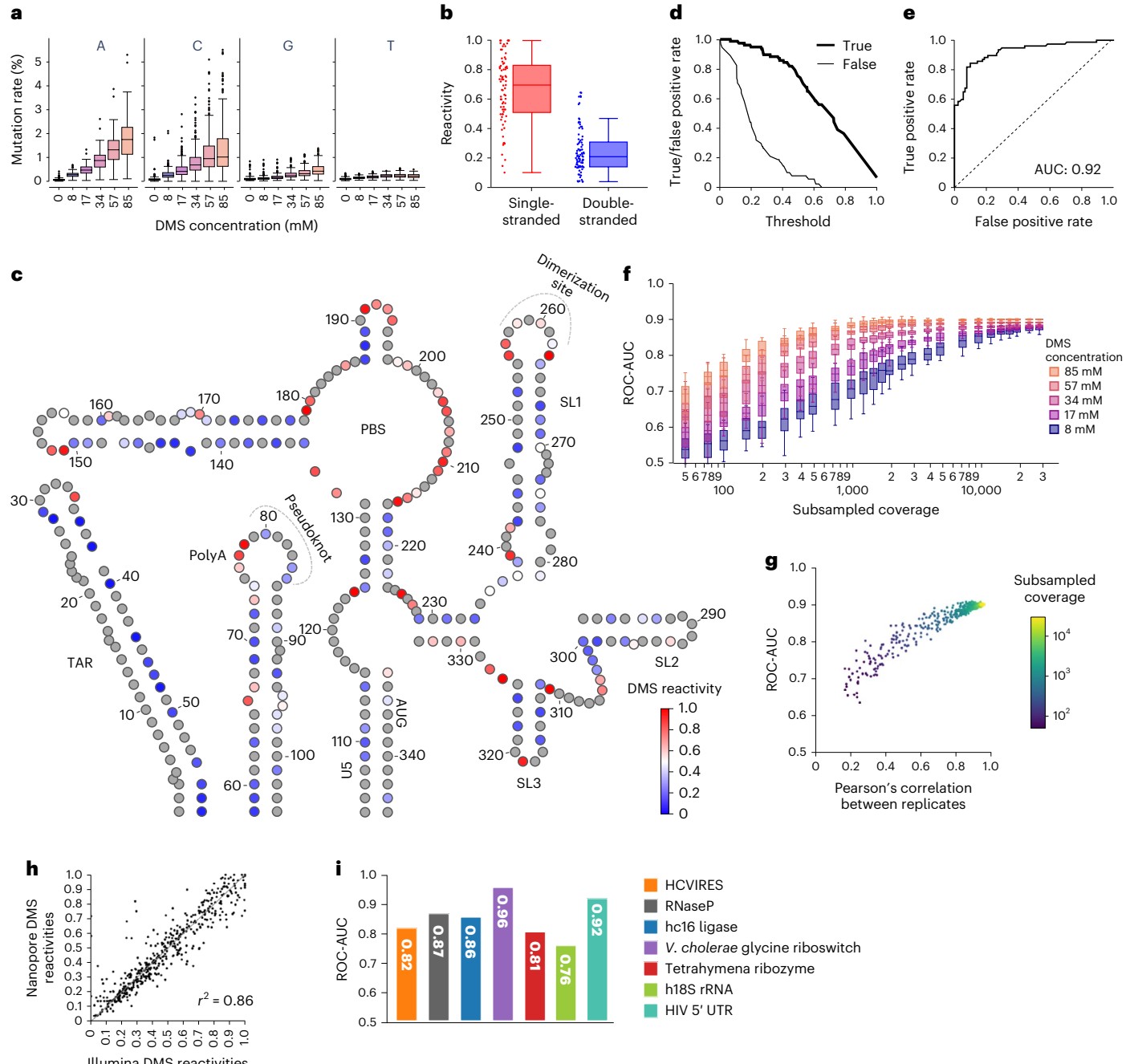

**Fig. 2 | DMS signals are detected and recapitulate the HIV-1 5′ UTR structure in the unspliced RNA.** Nano-DMS-MaP recapitulates the consensus structure of the HIV-1 5′ UTR of the unspliced RNA found in cells for two biologically independent replicates. **a**, Nucleotide-specific error rates per position as detected by rfcount using optimized settings at different DMS concentrations. **b**, Boxplot analysis of DMS reactivities for double- and single-stranded nucleotides. **c**, Consensus structure of the HIV-1 5′ UTR. A and C residues are colored according to DMS reactivities obtained at 85 mM using a blue-white-red color scheme. G and U are colored in gray. **d**, Curve representing true and false positive rates for a binary classifier for strandedness as DMS reactivity threshold is varied. **e**, ROC curve representing the false/true positive rates of a binary classifier for strandedness as the DMS reactivity threshold is varied. **f**, ROC-AUC scores of the match between mean DMS reactivities and the consensus structure for each DMS concentration as coverage is varied via subsampling ($n$ = 20). **g**, Plot showing the relationship between ROC-AUC scores and the Pearson's correlation between two replicates of the US RNA at 85 mM DMS concentration at different subsampling depths (indicated by color). **h**, Correlation between Nanopore and Illumina-generated DMS reactivities for matched US RNA probed at 85 mM. Coefficient of determination ($r^2$) is shown. **i**, ROC-AUC scores for mean Nano-DMS-MaP reactivity after probing at 57 mM DMS concentration for different in vitro transcribed RNAs, the human 18 S ribosomal rRNA and HIV 5′ UTR in cell. For all boxplots, boxes represent quartile 1 (Q1) to quartile 3 (Q3). The second quartile (Q2) is marked by a line inside the box. Whiskers correspond to the box edges ± 1.5× interquartile range (IQR) (Q3-Q1).

A comparison of the DMS reactivities across the entire HIV-1 genome showed a slight but consistent trend for increased DMS accessibility in the virions (Fig. 3a). The molecular basis for this is unknown, but relaxation of RNA structure in the virion may facilitate reverse transcription during the next cycle of replication. A notable exception to this trend was the 5′ UTR, which showed decreased reactivity to DMS in the virion compared with the cells (Fig. 3a). This can be explained largely by DMS reactivity changes due to annealing of the tRNA primer,

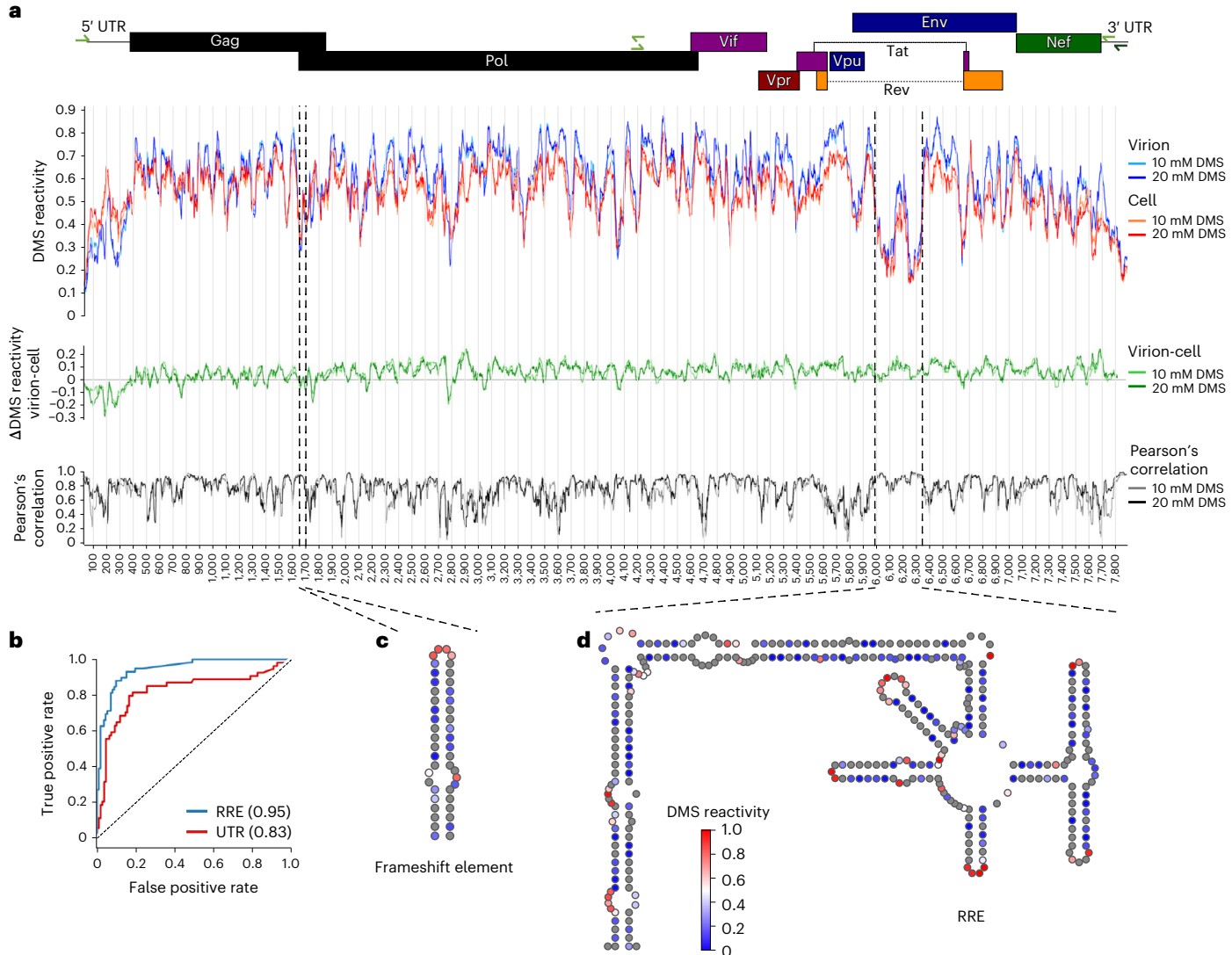

**Fig. 3 | In situ structure of HIV-1 genome in cells and virions.** Structure of the US HIV-1 RNA probed in situ by adding DMS directly to infected cells or viral supernatants. **a**, DMS reactivities were obtained for the near full-length genome from two overlapping amplicons of approximately 4 kb. Virion (blue) and cell (red) DMS reactivities shown are smoothed over a dynamic window containing 20 A or C nucleotides. DMS reactivity difference between virion and cell is shown in green (mean virion RNA – mean cell RNA). Pearson's correlations of the 20-nt window between the virion and cell DMS reactivities are shown in black. Genome position is shown on the *x* axis and the position of the frameshift element and the RRE is highlighted with dotted lines. Positions of HIV-1 ORFs are represented as boxes at the top of the plot. **b**, ROC curve representing the false/true positive rates of a binary classifier for strandedness as the DMS reactivity threshold is varied for the RRE and the HIV-1 5′ UTR. **c,d**, Structure of the HIV-1 frameshift element (**c**) and the RRE (**d**). A and C residues are colored according to DMS reactivities obtained at 20 mM using a blue-white-red color scheme. Blue, 0; white, 0.5; red, 1. Red nucleotides are reactive to DMS, indicating single strandedness. Blue nucleotides are unreactive to DMS, indicating double-strandedness or protection from modification due to occlusion, for example, by other protein or nucleic acid molecules. G and U are in gray.

dimerization at the apical loop of SL1 and potential binding sites for the viral nucleocapsid protein, NCp7 (Supplementary Fig. 2). Measured DMS reactivities correlated with known highly structured regions of the HIV-1 genome, such as the 5′ UTR (Fig. 3b), the two-helix model of the frameshift site (Fig. 3c), and the Rev response element (RRE) (Fig. 3b,d). Altogether, these data demonstrate the suitability of Nano-DMS-MaP for long-read structural probing.

**Long-read sequencing detects diverse transcript isoforms**

The HIV-1 genome is transcribed by the host cell into three major transcript classes: US, partially spliced (PS) and fully spliced (FS) (Fig. 4a). During the late stages of infection, HIV-1 specifically packages the US genome into viral particles. However, the PS and FS viral RNAs are efficiently excluded from the packaging process by a poorly understood mechanism. This mechanism of exclusion applies to the over 50 different spliced transcripts produced in HIV-1 infected cells through the use of a variety of weak donor and acceptor sites[32–35]. All viral RNAs share the first 289 nt of the 5′ UTR, including a major packaging signal, known as stem-loop 1 (SL1). SL1 contains a palindromic dimerization initiation sequence (DIS) within its apical loop, and is the primary binding site for the viral structural protein Pr55[Gag] (Fig. 4b)[36,37]. SL1 is included in all spliced viral RNAs because it lies upstream of the major splice donor site within stem-loop 2 (SL2) (Fig. 4b). Nevertheless, it has been reported that SL1 directs US, but not PS or FS, transcripts into nascent virions[27,36,38]. We therefore hypothesized that structural differences within the packaging signal shared between US and spliced transcripts may explain the selective recognition of the US transcripts by the viral packaging machinery. To address this hypothesis, we designed

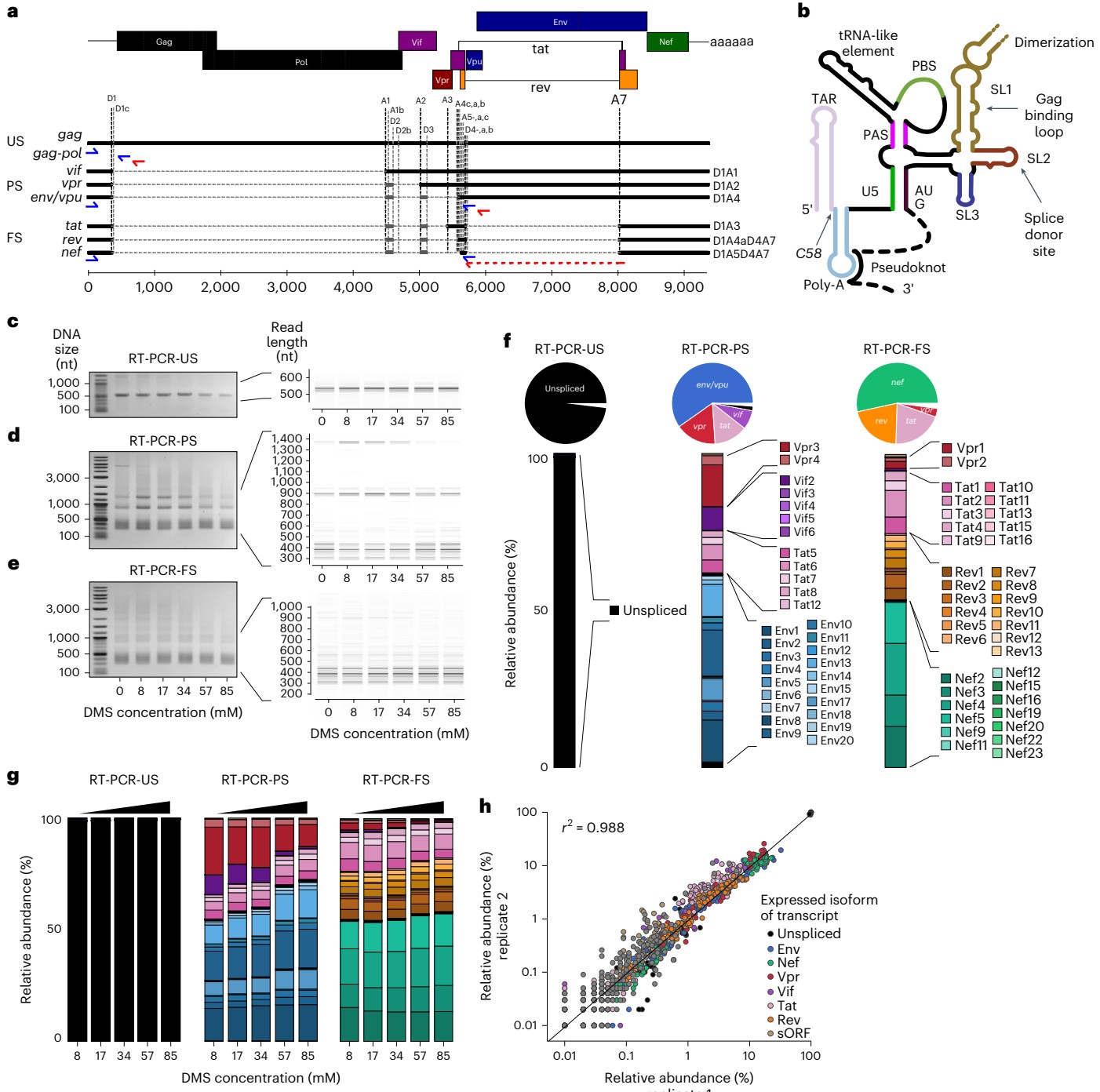

**Fig. 4 | Nanopore sequencing recovers a diverse isoform landscape.** Long-read nanopore sequencing identifies diverse transcript isoforms. **a**, HIV-1 has a complex splicing landscape due to the presence of major and minor splice acceptor sites within its genome. Three main classes of RNA are described: US, PS and FS. Specific RT primers for the US, PS and FS are shown in red, and amplification primers in blue. For the full genome, the specific RT primer is shown in dark green and the amplification primers in light green. **b**, Schematic of the HIV-1 5′ UTR of the US RNA. Major structural and function domains are indicated by bold lines. Other important features are labeled. The pseudoknot

and U5–AUG interaction is present only in the US RNA. **c**–**e**, Agarose and virtual gels showing diverse species amplified at each DMS concentration by specific RT-PCRs for US (**c**), PS (**d**) and FS (**e**) transcripts. Marker, 1 kb plus (NEB). **f**, Isoform-specific mapping detects many transcript isoforms. Transcript isoforms are colored by expressed viral protein. **g**, Relative abundance of transcript isoforms at each DMS concentration. **h**, Correlation between relative abundance of each transcript isoform between two independent experimental replicates for all samples (with and without DMS modification).

---

an experiment to sequence and detect the individual US, FS and PS RNAs. We then performed isoform-resolved RNA structure analysis.

We first established RT-PCR conditions specific for amplifying complex mixtures of US, PS and FS RNAs (Fig. 4c–e and Supplementary Fig. 3a–c). For the reverse transcription of the US RNA, we used an

RT primer that hybridized within the *gag* open reading frame (ORF). For the PS RNAs we used an RT primer complementary to a region in the D4–A7 intron (Fig. 4a), and for the FS RNAs we used an RT primer spanning the D4–A7 splice site (Fig. 4a). To PCR amplify the resulting DNA, we used PrimeSTAR GXL because we found it was able to amplify

long and diverse transcripts (Supplementary Fig. 3d). As expected, we detected a single 0.5 kb product for the US-cDNA (Fig. 4c). From the PS cDNA, we detected a variety of spliced transcripts (300–1.4 kb), as well as a transcript of around 5.5 kb corresponding to the US RNA (Fig. 4d). For the FS cDNA, we detected a different subset of spliced transcripts (300–900 bp) (Fig. 4e). PCR amplicons from the US, PS and FS samples were then barcoded and sequenced on the Oxford Nanopore Technologies MinIon device using kit v.12 (Q20+) chemistry. From four runs on four flow cells, we obtained 5 million demultiplexed reads (2.6 Gb) with a mean Q-score of 15.9 (97.4% accuracy). When sequenced reads were plotted as a virtual gel, the relative proportion and lengths of these reads correlated with species previously detected on agarose gels (Fig. 4c–e). The sole exception was the 5.5 kb transcript, which was readily visible on the agarose gel, but was present at much lower intensity on the virtual gel based on the nanopore sequencing reads (Fig. 4d and Supplementary Fig. 3e). These data indicate that nanopore cDNA sequencing can capture diverse transcripts, although there may be a bias against longer transcripts in complex mixtures arising either during library preparation and/or sequencing.

We next mapped individual reads from the untreated sample to the HIV-1 transcriptome using IsoQuant[39] (Methods). Across all samples, over 80% of reads were unambiguously assigned to a single known isoform (Supplementary Fig. 4a), showing efficient read-to-isoform assignment even in this complex splicing landscape. Approximately 13% of reads could be assigned equally well to several isoforms and were ignored in subsequent analysis (Supplementary Fig. 4a). In addition, 4% of reads were discarded because they could not be assigned to any known spliced isoform (Supplementary Fig. 4a). Sequencing reads from the US sample mapped almost exclusively to the genomic RNA (98%) (Fig. 4f,g). In contrast, reads from the PS and FS reactions could be assigned uniquely to many different spliced transcripts (Fig. 4f,g). In the PS sample, we identified 16 transcript isoforms with at least 1,000-fold coverage in both replicates (6 with 4,000-fold coverage at 57 mM DMS concentration), including transcripts expressing Env/Vpu, Tat, Vif and Vpr (Supplementary Fig. 4b). In the FS sample, we detected 15 transcripts with at least 1,000-fold coverage in both replicates (10 with 4,000-fold at 57 mM DMS concentration) which mapped to Nef, Rev, Tat and Vpr-expressing isoforms (Supplementary Fig. 4b). The most common splice acceptor site in PS transcripts was A1 (one-third of transcripts), found for example in Env13, Env5, Tat6 and Vif2 isoforms, followed by A2, found in Env9 and Vpr3 and A5, solely from Env1 expression. The general occurrence of acceptor sites was similar in FS transcripts (that is, D4A7-spliced). Here A1, found in Nef5, Nef3 and Tat2 isoforms, was the most common acceptor, followed by A2, found in Nef4, Rev7, Rev8 and Tat3, and A5, expressing the Nef2 isoform (Supplementary Fig. 4c). The relative quantities of recovered transcripts obtained by Nano-DMS-MaP were also highly reproducible ($r^2 = 0.988$) across two independent experiments (Fig. 4h). All transcripts we detected were seen in previous studies of HeLa cells expressing HIV-1 (refs. 33,34,40). In the presence of DMS, we observed a progressive decrease in the proportion of reads mapping to longer transcripts with increasing DMS concentration. In particular, the 1.4 kb Vif2 and 897 bp Vpr3 transcripts in the PS sample were decreased compared with the shorter Env-expressing transcripts (Fig. 4g). Nevertheless, the effects were modest at most of the DMS concentrations. Altogether, these data confirm that nanopore cDNA sequencing accurately captures a comprehensive and biologically relevant view of the HIV-1 splicing landscape.

### HIV-1 transcript isoforms have distinct structures
We next investigated the 5′ UTR structures of 16 different PS and FS HIV-1 transcripts in cells where read depths were more than 4,000 reads in both replicates, thus ensuring an informative and reproducible structural signal (mean Pearson's $r = 0.95$ at 57 mM) (Extended Data Fig. 9a). We also analyzed the native structure of the US RNA in virions.

A correlation analysis and hierarchical clustering of the DMS reactivity of the shared 5′ UTR of all isoforms revealed that the US RNA from cells and virions clustered together (Fig. 5a). As shown before (Fig. 3a), the tRNA primer binding site (PBS) and the dimerization initiation site (DIS) became fully protected from DMS in the virion, due to known intermolecular RNA interactions at these sites (Extended Data Fig. 9b). On the other hand, spliced viral RNAs had distinct, yet similar structural profiles, as they grouped together into their own cluster (Fig. 5a). Subclustering of the spliced RNAs was associated mainly with the first splice acceptor site usage, suggesting an effect of the sequence of the first adjacent exon on the 5′ UTR structure (Fig. 5a). To characterize this effect in more detail, we averaged the DMS reactivities according to first acceptor site (Fig. 5b). By subtracting spliced DMS reactivities from those of the US RNA we identified several regions within spliced transcripts showing strong and consistent changes in DMS reactivities indicative of structural rearrangements compared to the US RNA (Fig. 5b and Extended Data Fig. 9b,c). Increases in DMS reactivities at positions C80, C84 and C85 in the poly(A) loop, and positions C109, C110 and C111 in the U5 region, are likely explained by the loss of sequences that are present only in the US RNA (Fig. 4b and Fig. 5b). Increases in DMS reactivity at position C58 of the poly(A) stem cannot be explained by loss of downstream sequences, but may instead relate to transcription start site variation shown to regulate the translation and packaging of the US RNA via 5′ UTR remodeling[41,42] (Fig. 4b and Fig. 5b). The PBS was structured similarly in US and spliced RNAs, although there were distinct changes in the PAS stem, such as an increase in reactivity at position A220 and decreases in reactivity at A225 and A227 (Fig. 5b). Most strikingly, we observed clear increases in reactivity throughout the 3′ portion of the SL1 stem, indicating its structural reorganization (Fig. 5b).

With our isoform-specific probing data we next performed de novo folding of the individual isoforms. This analysis confirmed structural rearrangements associated with the first splice acceptor usage, but more generally an unfolding of several regions within the spliced RNA (Supplementary Data Files 2 and Extended Data Fig. 9d). Specifically, whereas the transactivation repeat (TAR) stem-loop was found in all isoforms, the poly(A) and tRNA-like structures were predicted to form only in D1A1/A2-spliced isoforms. The SL1 stem-loop, which is bound by the viral Pr55^Gag protein during packaging[36,37], was never predicted as its canonical stem-loop (Fig. 5c,d and Extended Data Fig. 9d). Instead, for the D1A1/2 spliced isoforms, we identified an anti-PAS-SL1 interaction reported to promote the structural rearrangement of the HIV-1 5′ UTR of NL4-3 RNA into a monomeric and packaging incompetent conformation[9] (Fig. 5d). Taken together, these data show that the 5′ UTR is remodeled upon removal of intron 1, leading to structural reorganization of a packaging motif, which likely explains the exclusion of spliced RNA from the virion.

## Discussion
Nano-DMS-MaP is a rapid, reproducible and straightforward method for long-read and isoform-resolved RNA structural probing. Using an ultraprocessive reverse transcriptase, we were able to generate long cDNA molecules with mutational signatures at sites of DMS modification and showed that nanopore cDNA sequencing can be used for RNA structure determination by mutational profiling. Nano-DMS-MaP therefore enables the identification of new regulatory mechanisms that are hidden in short-read ensemble analyses.

Despite the high intrinsic error rates of the nanopore sequencing platform, we were able to recover structural information. Critically, we found that ignoring indels during mutation counting decreased the substitution error rates by an order of magnitude. Together with additional quality score filters, we achieved an effective accuracy of 99.9% in the untreated control for single point mutations. This is equivalent to a Phred quality score of 30, which is widely considered a benchmark accuracy in next-generation sequencing[43,44]. This was

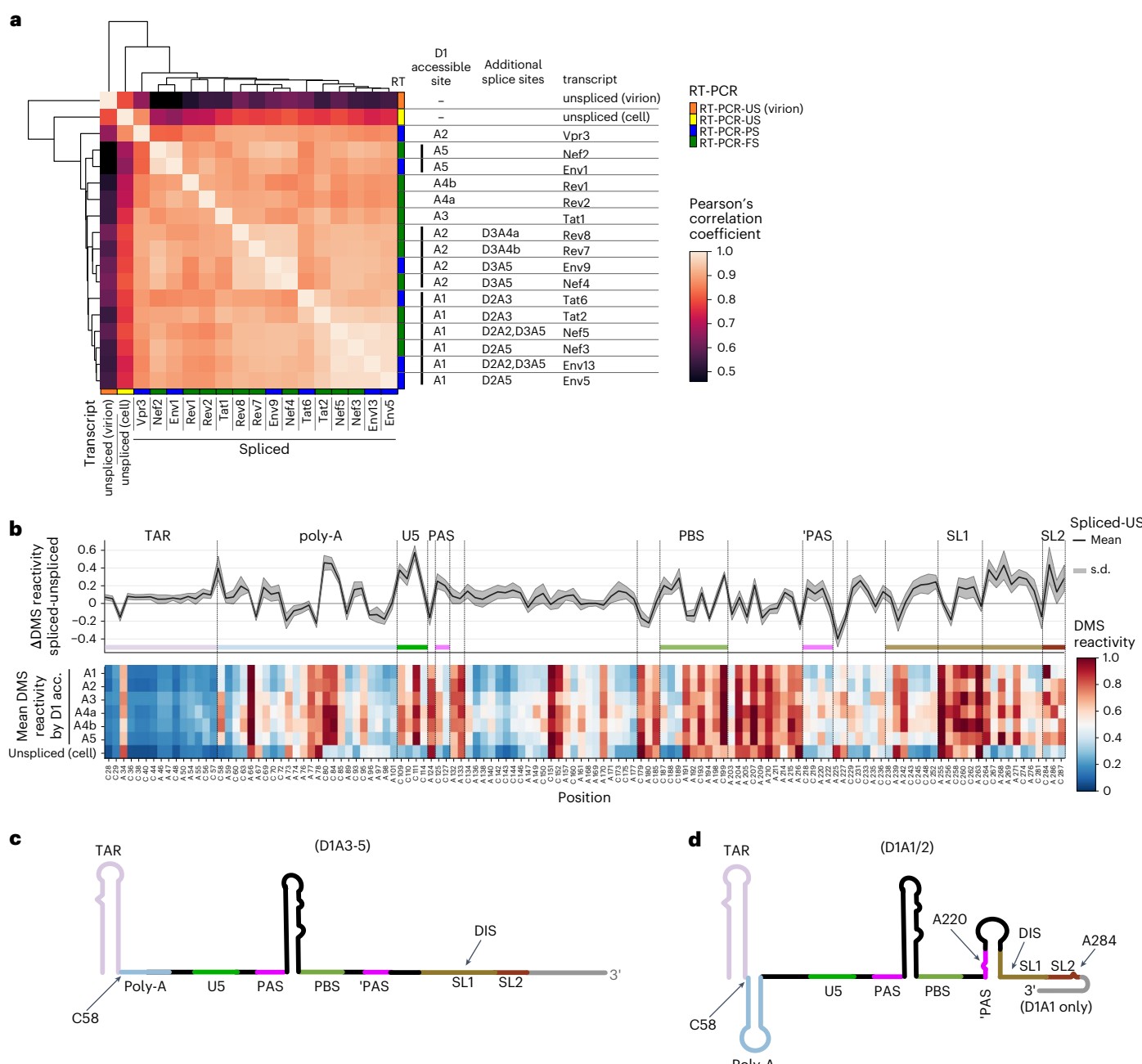

**Fig. 5 | Isoform-resolved in cell and in virion probing reveals distinct alterations in 5′ UTR structure of spliced mRNA.** Spliced and US HIV-1 transcripts have distinct structures in the HIV-1 5′ UTR. **a**, Hierarchical clustering heatmap of DMS reactivities. Correlations of the shared first 289 nt of native cellular RNA probed with 57 mM mean DMS reactivity of A and C residues for two biologically independent replicates are calculated using Pearson's correlation coefficient and hierarchical clustering using the nearest point algorithm. Transcript and spliced acceptor site usage is indicated. **b**, Upper plot, per position ΔDMS reactivity (mean spliced RNA − unspliced RNA).

Mean, black line; s.d., gray shading. Lower plot, heatmap of DMS reactivities at A and C residues for spliced and US RNA using a blue-white-red color scheme. Blue, 0; white, 0.5; red, 1. **c**–**d**, Structural models of the HIV-1 spliced RNAs inferred from de novo structure prediction (Extended Data Fig. 9) within important structure and functional domains labeled. D1A3-5 spliced viral RNAs are unstructured except for the TAR stem-loop and the tRNA-like element (**c**), whereas D1A1/2 spliced viral RNAs additionally contain the polyA stem-loop and an anti-PAS-SL1 interaction (**d**).

made possible because Nanopore datasets have unique error profiles with higher likelihood of indels compared to single point mutations (Supplementary Fig. 5a). Ignoring indels did not result in lower quality structural data because MarathonRT almost always introduces single nucleotide mutations from DMS modifications. This allowed us to separate the DMS signal from the background noise introduced from nanopore basecalling errors (Extended Data Fig. 4b,c). Other commercially available highly processive RTases,

such as TGIRT-III, have similar characteristics on DMS-modified RNAs and may also be appropriate for Nano-DMS-MaP[12,45]. In agreement with previous reports, we also found that DMS can provide valuable structural information at U residues (Extended Data Fig. 10a–f)[46,47]. Thus, where read depths and DMS concentrations are high, information at U residues may be cautiously included in RNA structure analysis (Extended Data Fig. 10g). Mutations at G residues, however, did not correlate with secondary structure. This is because methylation at G

residues occurs preferentially at the N7 position on the Hoogsteen-face, leading to a characteristic G to A substitution with MarathonRT (Extended Data Fig. 4b,c)[5]. Instead, this signal could be used to study noncanonical RNA structures involving Hoogsteen interactions, such as G-quadruplexes[48,49].

Our method is related to recent advances in long-read RNA structural probing by nanopore direct RNA sequencing (dRNA-seq) of chemically modified transcripts[50–52]. However, dRNA-seq structural probing requires specialized modification detection algorithms which may need continual updates as nanopore sequencing chemistry changes. Nano-DMS-MaP on the other hand immediately takes advantage of improvements in DNA basecaller accuracy, leading to higher signal-to-noise ratio without changes in the experimental or analytical pipeline. Moreover, in comparison with direct RNA sequencing, nanopore DNA sequencing has a higher throughput on the same flow cells, resulting in a lower cost per base than equivalent methods using dRNA sequencing.

Using Nano-DMS-MaP, we recovered high quality structural information on RNA molecules up to 4 kb in length, which would allow isoform-resolved analysis of most human mRNAs[53]. An important caveat is that we used lower DMS concentrations than typically used in short-read DMS-MaP experiments. Although the accuracy of Nano-DMS-MaP structural information was similar at low and high DMS concentrations, higher read depths are generally required for longer molecules (Fig. 2g). For example, the 1.5 kb long RNA only required 4,000-fold coverage (6 megabases (Mb)), but the 4 kb long RNA required 20,000-fold coverage (80 Mb). Thus, sequencing throughput can be limiting, especially when using lower DMS concentrations to analyze longer transcripts. Additionally, when sequencing complex mixtures, it may not always be possible to structurally characterize transcript isoforms of low abundance due to the small number of reads captured. Future increases in nanopore sequencing throughput, together with selective sequencing, may alleviate these limitations. Alternatively, further improvements in reverse transcriptase processivity would allow higher modification densities without the same tradeoffs in read length, which would reduce sequencing requirements to eventually deliver transcriptome-wide structural probing of RNA isoforms. Nano-DMS-MaP may also in time allow more accurate structural determination through the detection of long-range interactions by correlated chemical probing[7,9,31,54] and computational deconvolution of structural ensembles[8,55–57]. Favorable characteristics of Nano-DMS-MaP for these analyses are a higher number of mutations per read and a low background nucleotide substitution rate compared with equivalent Illumina based methods (Extended Data Fig. 4 and Supplementary Fig. 5).

By applying our method to the complex splicing landscape of HIV-1 we identified strong and consistent increases in DMS reactivity in the SL1 stem of spliced viral RNAs, which is a structure involved in Pr55$^{Gag}$ binding[27,36,37]. DMS-guided structural predictions revealed restructuring of SL1, although the underlying mechanism is unclear. One possibility is that the loss of the SL2 hairpin containing the splice donor site indirectly destabilizes SL1 structure[9]. Alternatively, sequences downstream of the splice site may be required to fold the HIV-1 5′ UTR into a packaging competent structure[36]. In support of the second possibility, DMS reactivity changes in the spliced RNAs clearly show the loss of the U5–AUG interaction[58] and a pseudoknot interaction between the poly(A) stem-loop and sequences in *gag*[59]. The U5–AUG[27,60–62] and polyA[9,63] structure have both been implicated in 5′ UTR structural switching and the selective packaging of the US RNA[9,63]. We also observed increased DMS reactivity changes at position C58 that is linked to transcription start site variation that alters 5′ UTR structure to regulate genome packaging and translation of the US RNA[42,64,65]. We also cannot exclude that the unfolding of SL1 is due to the preferential translation of the spliced viral RNAs themselves. Testing whether SL1 unfolding drives RNA packaging selectivity in cells,

and the role of transcription start site variation on 5′ UTR folding and translation are key topics for future studies.

## Online content

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

## Methods

### Cell culture and virus production

HEK 293T cells were obtained from the American Type Culture Collection and maintained in high glucose Dulbecco's modified Eagle's medium with GlutaMAX (DMEM; Gibco) supplemented with 10% (v/v) iron supplemented calf serum (Sigma-Aldrich). Cellular and viral RNA was produced by reverse transfection of 10 million cells with 3 µg of pDRNL43Δenv[66] and 36 µl of 1 mg ml⁻¹ transfection grade linear polyethylenimine (PEI MAX MW 40,000; Polysciences), following the manufacturer's recommendations.

### DMS probing and RNA extraction

At 24 h posttransfection, DMS probing of viral RNA was carried out under native conditions from cells and viruses. Viral particles were collected as 15 ml viral supernatants, which were then clarified by centrifugation for 10 min at 5,000$g$ and subsequently passed through 0.45 µm filter to remove cellular debris. Purified virus was concentrated by ultracentrifugation at 100,000$g$ through a 20% sucrose cushion (w/v) in Tris-HCl 50 mM pH 7.4, NaCl 100 mM, EDTA 0.5 mM). After centrifugation, the viral pellet was gently resuspended in virus resuspension buffer (50 mM HEPES-KOH pH 8.0, 200 mM NaCl, 3 mM MgCl₂). Samples were probed by adding a one-tenth volume of DMS diluted to the correct concentration in ethanol directly to cells or viruses. DMS treated samples were incubated for 6 min at 37 °C and quenched with one-tenth volume of β-mercaptoethanol (Sigma-Aldrich). For each sample, equivalent control reactions were performed incubating samples for 6 min at 37 °C in the presence of one-tenth volume ethanol instead of DMS. Probed and control RNA samples were purified by extraction with Tri reagent LS (Sigma-Aldrich), according to the manufacturer's instructions with the addition of 1 µl of glycoblue (ThermoFisher Scientific) during isopropanol precipitation to enhance recovery of low abundance RNA from the viral samples. Purified RNA was resuspended in RNase-free H₂O. Then, 16 µg of cellular RNA and the totality of the viral RNA was treated with 3 µl Turbo DNase (ThermoFisher Scientific), 12 U of RNasin and 5 µl of 10× Turbo DNase buffer in a 50 µl volume for 30 min at 37 °C. Following DNase treatment, RNA was column-purified using NTC buffer and the NucleoSpin Gel and PCR Clean-up kit (Macherey-Nagel), according to the manufacturer's instructions.

### Preparation and probing of in vitro transcribed RNA

DNA templates of the bacterial RNase P type A[67], hc16 ligase, tetrahymena riboswitch and *Vibrio cholerae* glycine riboswitch[30] were generated by assembly from DNA oligos (IDT) according to the primerize scheme[68]. The HCV IRES was subcloned from a reporter plasmid kindly provided by N. Caliskan. Fully assembled products were cloned into pJet-1.2 vector (ThermoFisher Scientific) for propagation and confirmed by Sanger sequencing. DNA was then amplified from plasmids with primer T7_fw (AAAGAATTCTAATACGACTCACTATAGG) and the M13_pA_re (TTTTTTTTTTTGATTATCATACTCTGATAATCCAGGAAACAGCTATGACCATG) with the exception of HCV IRES, for which primer T7_HCV_fw (AAAGAAGACTTGGGGTAATACGACTCACTATAGGCCAGCCCCCGATTG) was used. DNA amplicons were then purified with 1.2× SPRI bead purification with Mag-Bind TotalPure NGS beads (Omega Biotek). Briefly, the DNA-bead mixture was incubated for 5 min under light agitation and beads were pelleted on a magnetic rack (Invitrogen DYNAL), followed by removal of supernatant and two washes with 100 µl freshly prepared 70% ethanol. Finally, beads were air-dried for 3–5 min (until appearance changed from glossy to rough) and DNA was eluted by addition of 15 µl H₂O, followed by 5 min incubation at room temperature.

For in vitro transcription 500 fmol of DNA were prepared in 40 mM Tris pH 7.5, 18 mM MgCl₂, 10 mM DDT, 1 mM spermidine, 5 mM NTPs, 40 U RNasin (Molox) and homemade T7 RNA polymerase for 3 h at 37 °C, followed by DNase I treatment for 30 min at 37 C and 1.6× SPRI bead purification.

For probing, 300 ng of the RNA was prepared in an 8 µl reaction mix in 0.5 mM EDTA, 30 mM HEPES pH 7.5, 300 mM NaCl and heated to 95 °C for 1 min, followed by placing on ice. To facilitate folding of the RNA 1 µl of 50 mM MgCl₂ (5 mM final concentration) was added before incubation at 37 °C for 30 min. To probe the RNA, 1 µl DMS diluted in ethanol was added at the indicated final concentrations before incubation at 37 °C for 7 min. The reaction was quenched with four volumes of 30% β-mercaptoethanol. RNAs probed at the same concentration were pooled, 0.1 volume of 3 M NaOAc and 3 volumes EtOH were added, and RNA was precipitated at −20 °C overnight. RNA was then pelleted by centrifugation at 16,000$g$ for 30 min, washed twice with 70% EtOH before resuspension in H₂O and normalization to 100 ng µl⁻¹.

### Reverse transcription

Reverse transcription was performed on 1 µg of cellular RNA, the totality of viral RNA or 300 ng of purified in vitro transcribed RNA using MarathonRT[29]. pET-6×His-SUMO-MarathonRT encoding MarathonRT was a gift from A. Pyle (Addgene plasmid catalog no. 109029; http://n2t.net/addgene:109029; RRID: Addgene_109029), and the enzyme was purified according to Zhao et al.[29]. US HIV-1 RNA was reverse transcribed using primer RT-US (GATGGTTGTAGCTGTCCCAGTATTTGTC), PS RNA using primer RT-PS (CTCCTTCACTCTCATTGCCACTGTC) and FS using primer RT-FS (CTCGGGGTTGGGAGGTGGGTTGC). The full-length genome was reverse transcribed with RT-FL (GAAGCACTCAAGGCAAGC). Human 18S rRNA was reverse transcribed with primer RT-h18S (TAATGATCCTTCCGCAGGTTCACCTAC), and the in vitro transcribed RNAs were reverse transcribed with primer M13_pA_re.

RNA was first mixed with 0.5 mM dNTPs, 50 nM of primer brought to 9 µl with RNase-free H₂O and denatured for 5 min at 65 °C. Samples were placed on ice for 2 min and reverse transcription was initiated by adding 40 U of MarathonRT in 50 mM Tris-HCl pH 8.3, 200 mM KCl, 20% glycerol (v/v), 0.4 mM MnCl₂, 4 U of RNasin in 20 µl total volume. Samples were incubated for 4–8 h at 55 °C for primer RT-US or 42 °C for primers RT-PS, RT-FS and full-length. No reverse transcriptase controls were carried out as above, with the omission of the MarathonRT enzyme.

### PCR amplification of viral spliced and unspliced HIV-1 RNAs

RT reactions were diluted one to eight to a total volume of 160 µl with nuclease free H₂O. Viral RNA species were differentially amplified using primer pairs PCR-HIV_Fw (GGTCTCTCTGGTTAGACCAGATCTGAG) and PCR-US_Rv (GATGGTTGTAGCTGTCCCAGTATTTGTC) for US RNA, PCR-HIV_Fw and PCR-S_Rv (TTCGTCGCTGTCTCCGCTTC) for PS and FS RNA, PCR-HIV_Fw and PCR-A_Rv (CCCTGTCTCTGCTGGAATTACTTC) or PCR-B Fw (GAAGTAATTCCAGCAGAGACAGGG) and PCR-B-Rv (GAAGCACTCAAGGCAAGCTTTATTG) for the full-length genome. For h18S amplification primers h18S_fw (TACCTGGTTGATCCTGCCAGTAGCATATG) and h18S_re (TAATGATCCTTCCGCAGGTTCACCTAC) were used, whereas for the in vitro transcribed RNAs RNA-specific forward primers (Supplementary Table 1) were used together with primer M13_pA_re. PCR amplification conditions were 5 µl of diluted RT reaction with 0.05 U of PrimeSTAR GXL polymerase (Takara Bioscience), 250 nM of each primer, 200 µM of each dNTP and 1× PrimerSTAR GXL buffer in a total volume of 50 µl. Cycling conditions were initial denaturation for 2 min at 98 °C, followed by 25 (IVT, h18S), 27 (US) or 31 (PS, FS) cycles for 10 s at 98 °C, 15 s at 55 °C and 0.6 min (US), 0.75 min (IVT), 1 min (h18S) or 5.6 min (PS, FS) at 68 °C, followed by a final extension for 7 min at 68 °C. For the full-length genome amplification, we performed similar cycling conditions with 28 cycles, and we used an extension time of 20 s kb⁻¹ (1.5 min total). Amplicon quality was checked on 1% agarose gel poststained in EtBr.

### Nanopore sequencing

DNA amplicons were purified via SPRI bead purification by addition of 0.7× volumes of beads for US, PS, FS, full-length or 1.2× volumes for IVT and h18S samples as described above. Next, 25–80 ng of DNA

of each sample in volume of 5 µl underwent simultaneous dA-tailing and 5′-phosphorylation by addition of 0.7 µl NEBNext End-Repair Buffer and 0.3 µl NEBNext End-Repair enzymes followed by an incubation at room temperature for 5 min and 65 °C for 5 min. Barcodes of kit SQK-NBD112-96 (Oxford Nanopore Technologies) were then ligated in a 6 µl reaction containing 1 µl $H_2O$, 1 µl end-repaired DNA, 1 µl barcode and 3 µl NEB Blunt/TA Ligase Master Mix for 20 min at room temperature. Ligation was terminated by addition of 1 µl EDTA (SQK-NBD112-96), samples were pooled and purified with 0.4× Ampure XP beads (SQK-NBD112-96), and washed twice with short fragment buffer (SFB, SQK-NBD112-96). After elution of the pooled barcoded DNA in 35 µl $H_2O$, the motor protein was ligated in a 50 µl reaction containing 30 µl barcoded DNA, 10 µl NEBNext Quick Ligation Reaction Buffer (NEB B6058S), 5 µl AMII H (Oxford Nanopore Technologies SQK-NBD112-96) and 5 µl NEB T4 DNA ligase high concentration (NEB T2020M). Following a 20-min incubation at room temperature, the sample was purified with 0.4× Ampure XP beads and washed twice with SFB, taking care to not let the beads dry out between washes or before elution. The library was then sequenced on a R10.4 flow cell (FLO-MIN112, Oxford Nanopore Technologies) on a Minion Mk1B sequencer (Oxford Nanopore Technologies) using MinKnow acquisition software (Oxford Nanopore Technologies) v.21.11.8.

### Basecalling and isoform detection
Data was basecalled with guppy v.6.1.3 with the following parameters: '–do_read_splitting –c dna_r10.4._e8.1_sup.cfg –min_qscore 10 –barcode_kits SQK-NBD112-96 –trim_strategy dna –trim_barcodes –trim_adapters'. Virtual gels were generated with a custom python script using the numpy library, with intensity scaled by read length (that is, normalized by mass).

Read-to isoform mapping was performed using IsoQuant v.2.0 (ref. 39), using a general feature file (GFF) generated from previously published data (including transcript naming, as listed in Supplementary Table 2)[33], but adjusted for PCR primer start and end sites. Only reads mapping uniquely to one isoform were subsorted for subsequent analyses. Sorted reads were first aligned to their specific reference sequences using LAST v.1419, by first indexing the transcript reference with 'lastdb –uNEAR –R01', then training mismatch matrices per sample with 'last-train –Q0', followed by alignment with 'lastal -Qkeep –m20 –p {mismatch_matrix_file} | last-split –m1.' The output maf file was then converted to a Sam file with the 'maf-convert sam' command. The SAM file was then processed using Samtools v.1.12. Briefly, using Samtools a header was added, the file was converted to BAM format, followed by sorting the BAM file and lastly indexing it. The final BAM files were then used as input for the mutational profiling analysis. Mismatch types were analyzed from BAM alignment files with perbase v.0.8.3 and custom python scripts. Figures were generated using the python library plotly v.4.14.3.

### Mutational profiling analysis
Mutational Profiling analysis was performed for each isoform separately using RNAFramework v.2.7.2 (ref. 24) or custom python scripts. Mutation counting was performed with rfcount ('-mf mask_primers. csv –count-mutations –eq 10 –q 22 –mm –nd -ni') followed by reactivity normalization with rf-norm ('–scoreMethod Siegfried –normMethod 2 –reactiveBases all –maxUntreatedMut 0.05 –maxMutationRate 0.2 –norm-independent'). The signal-to-noise ratio was calculated per position as the mutation rate in DMS-probed sample divided by the mutation rate in the control as reported in the rfcount output files.

Calculating the correlation of reactivity between the two replicates was performed per sample and transcript with the RNAFramework tool rf-correlate for A and C residues only if not stated otherwise. Reactivity of biological replicates were combined for plotting and folding prediction via the rf-combine tool. ROC-AUC scores of reactivity profiles from the unspliced RNA in cells with increasing DMS concentration were calculated using a reference HIV NL4-3 5′ UTR structure[14] as ground truth data. Calculations were performed using the python package scikit-learn v.0.21.3.

Subsampling was performed on aligned BAM files of both replicates of the unspliced isoform in cells. First, the average coverage was determined with Samtools depth, which was then used to calculate the fraction of reads of the BAM file to be subsampled at each subsampling depth. Subsampled BAM files were then processed as described above, including the use of rf-combine to average the reactivities of both subsampled replicates before evaluating their ROC-AUC score.

### De novo RNA structure folding
De novo folding of isoform structures with coverage of at least 4,000 reads was performed by converting the reactivities on A and C residues to bp2seq files for input in EternaFold v.1.3.1 (ref. 69). The command to perform DMS-guided secondary structure prediction was 'eternafold predict {bp2seq_file} –evidence –numdatasources 1 –params EternaFold/parameters/EternaFoldParams_PLUS_POTENTIALS.v1'. Structures were then plotted using VARNA v.3-93 (ref. 70).

### Illumina sequencing
The amplicons from the US RNA sequenced previously by nanopore were prepared for Illumina sequencing as follows: The addition of the transposon sequence HF in 5′ and HR in 3′ of the amplicons was performed by five cycles of amplification using the primers PCR-HIV-HF_Fw (TCGTCGGCAGCGTCAGATGTGTATAAGAGACAGggtctctctggttagacc) and PCR-US-HR_Rv (GTCTCGTGGGCTCGGAGATGTGTATAAGAGACAGgatggttgtagctgtcccag) using the Q5 DNA Polymerase (NEB). DNA amplicons were purified via SPRI bead as described above using one volume of beads. Purified products (25 ng) were used in the final sequencing library preparation with Nextera DNA Flex Library Prep (Illumina) and Nextera DNA CD Indexes (96 indexes, 96 samples, Illumina), according to the manufacturer's instructions. Paired-end PE150 sequencing was carried out on an Illumina Novaseq instrument (Novogene). Fastq files were preprocessed with cutadapt v.4.1 with the following parameters '-a CTGTCTCTTATA -A CTGTCTCTTATA –nextseq-trim 25 –minimum-length 25 –max-n 0.' The trimmed files were then aligned using bowtie v.2 with parameters '-D 20 -R 3 -N 1 -L 15 -i S,1,0.50,' followed by the same analytical pipeline used for the nanopore data.

### Statistics and reproducibility
DMS reactivity data shown are the mean of two independent biological replicates of Nano-DMS-MaP experiments on A and C residues unless stated otherwise. RT-PCRs optimization experiments were performed at least twice.

### Reporting summary
Further information on research design is available in the Nature Portfolio Reporting Summary linked to this article.

## Data availability
Sequencing data is available at Sequence Read Archive (SRP424422, Bioproject ID PRJNA938445). DMS reactivities as csv files are provided as csv files in Supplementary Dataset 1. De novo predicted structures with Eternafold are provided as db and varna files in Supplementary Dataset 2. Source data are provided with this paper.

## Code availability
Code used for the Nano-DMS-MaP analysis is accessible via the Smyth-lab Github (https://github.com/smyth-lab/Nano-DMS-MaP).

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

## Acknowledgements

We would like to thank A. Sparmann for critical feedback. We also thank the Helmholtz Association (VH-NG-1347 to RS) and the Bundesministerium für Bildung und Forschung (BMBF) (COMPLS-182 to RS). A.-S.G.-B. was supported with a fellowship from the Peter und Traudl Engelhorn Stiftung and a Post Doc Plus funding (Graduate School of Life Sciences, University of Würzburg). U.B.A. was supported by a fellowship from the Deutschen Akademischen Austauschdienstes (DAAD). The funders had no role in study design, data collection and analysis, decision to publish or preparation of the manuscript.

## Author contributions

R.P.S., A.-S.G.-B. and P.B. conceived the study. A.-S.G.-B. performed the cellular probing experiments. P.B. performed the sequencing and in vitro probing experiments. P.B. and R.P.S. performed the analysis. U.B.A. performed cloning. R.P.S., P.B. and A.-S.G.-B. wrote the manuscript.

## Funding

## Competing interests

The authors declare no competing interests.

## Additional information

**Extended data** is available for this paper at https://doi.org/10.1038/s41592-023-01862-7.

**Correspondence and requests for materials** should be addressed to Redmond P. Smyth.

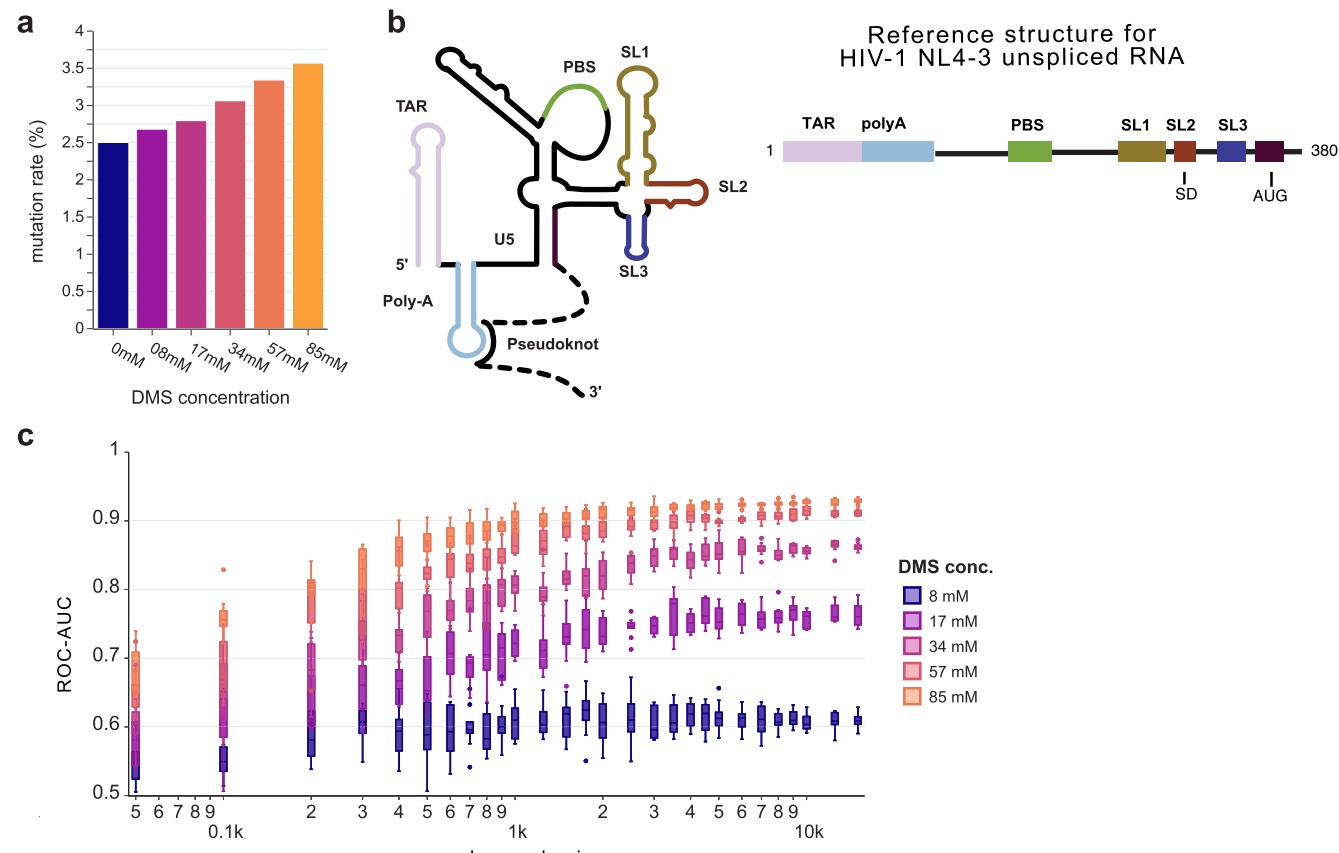

**Extended Data Fig. 1 | Mutation rates and ROC-AUC scores for unoptimized Nano-DMS-MaP-seq on the consensus structure of unspliced HIV-1 5'UTR.** (**a**) Representative nanopore sequencing error rates for different reverse transcription polymerase chain reaction (RT-PCR) at different DMS concentrations for the unspliced (US), single spliced (SS) and fully spliced (FS) samples from HIV-1 expressing cells. Also shown for the US RNA extracted from virions. (**b**) Structural model of the HIV-1 5' untranslated region (UTR). Major structural domains are indicated. Transactivation repeat (TAR), polyadenylation (polyA) stem loop, unique 5' regions (U5), primer binding site (PBS), stem loop 1 (SL1), stem loop 2 (SL2), stem loop 3 (SL3). A pseudoknot interaction with a downstream region (**c**) *Receiver Operating Characteristic Area Under the Curve (*ROC-AUC) scores indicating the match between mean DMS reactivities of two biological replicates and the consensus structure for each DMS concentration at different subsampling depths (n = 20). Boxes represent quartile 1 (Q1) to quartile 3 (Q3). The second quartile (Q2) is marked by a line inside the box. Whiskers correspond to the box' edges +/−1.5 times the interquartile range (IQR: Q3-Q1).*

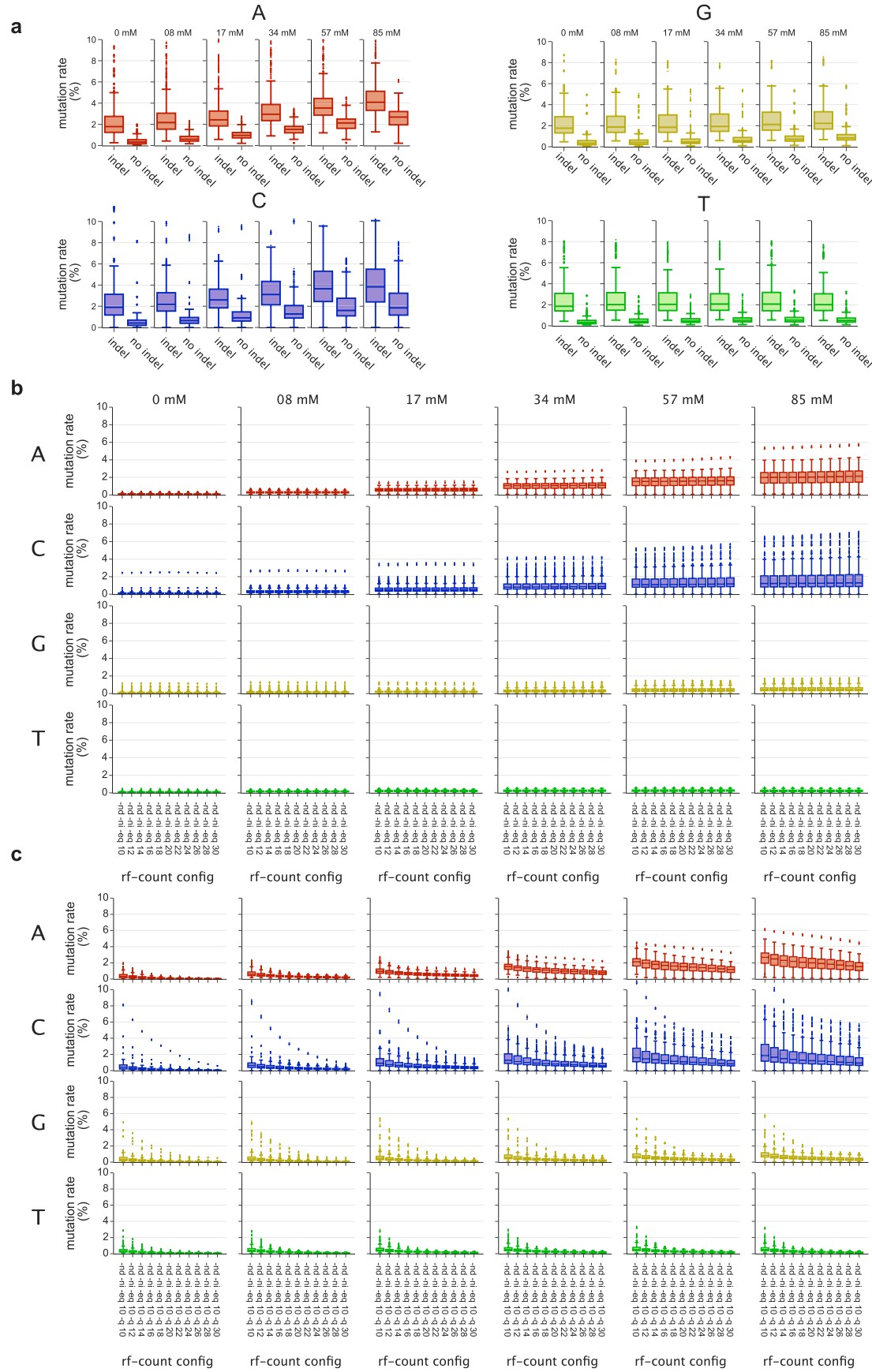

**Extended Data Fig. 2 | See next page for caption.**

**Extended Data Fig. 2 | Optimization of mutation counting.** Optimization of mutation counting based on one of the two experimental replicates. (**a**). Global mutation rates for unspliced HIV-1 RNA at each nucleotide for each DMS concentration, with and without counting insertions and deletions (indels). (**b**) With median read quality score filters of different stringencies.

(**c**) With position-wise quality score filters of different stringencies. *Boxes of (**a**–**c**) represent quartile 1 (Q1) to quartile 3 (Q3). A line inside the box marks the second quartile (Q2). Whiskers correspond to the box' edges +/–1.5 times the interquartile range (IQR: Q3-Q1).*

**a** Mean per read filter: Q10. Subsample size: 4000. **With indels.**

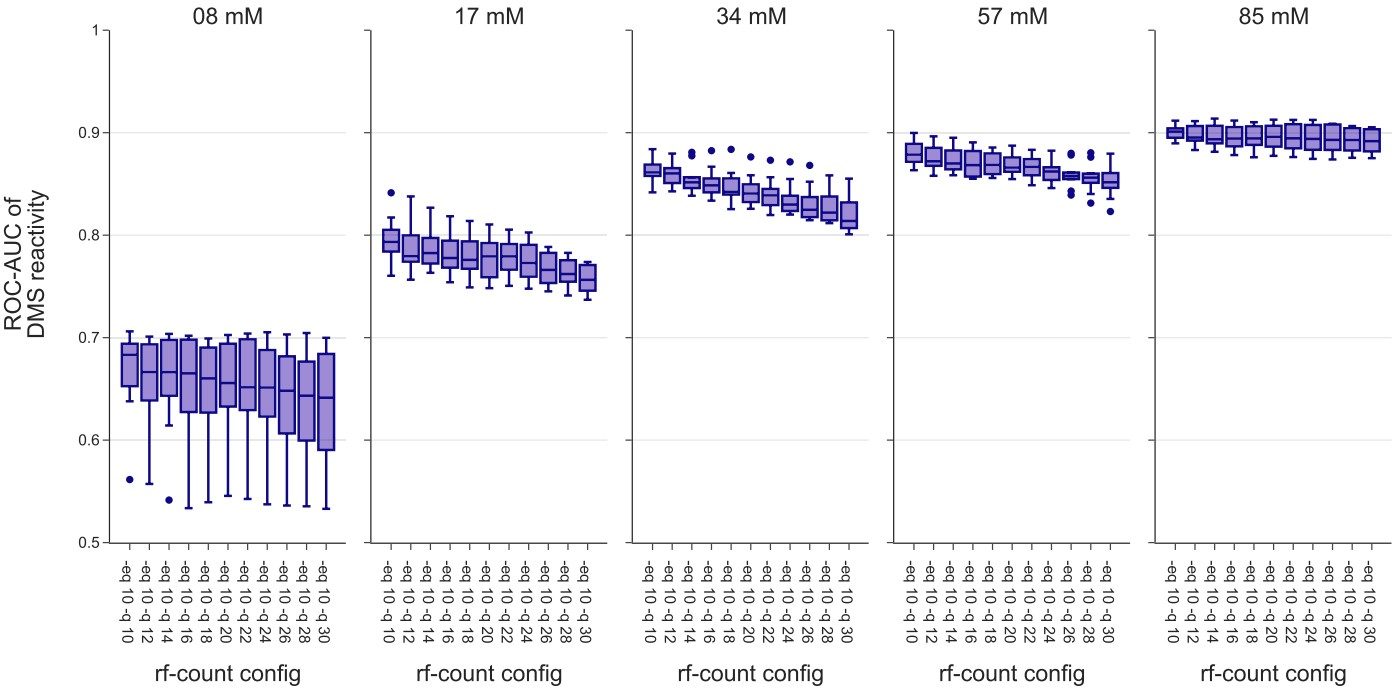

**b** Mean per read filter: Q10. Subsample size: 4000. **No indels.**

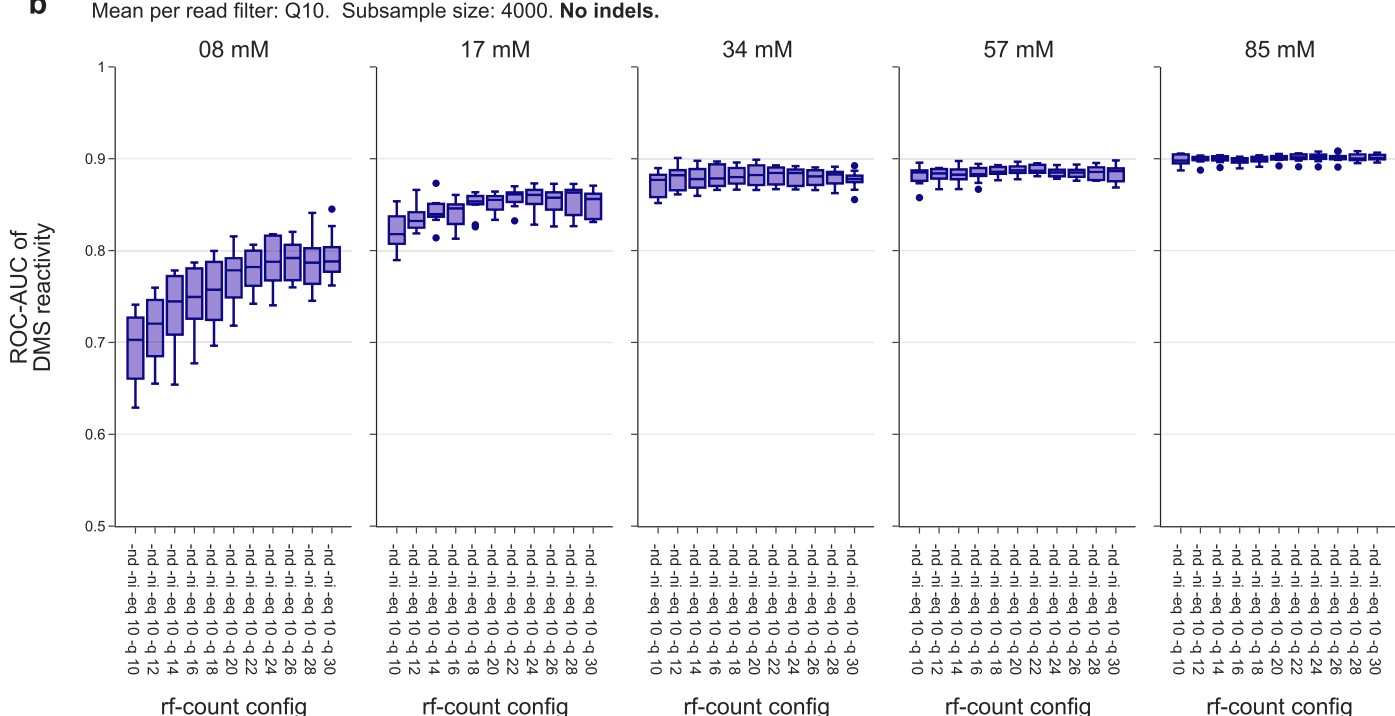

**Extended Data Fig. 3 | Optimization of ROC-AUC.** Effects of filtering settings on ROC-AUC scores, I.e., match between DMS reactivities and the consensus structure. ROC-AUC scores were calculated by sub-sampling of one replicate of the HIV 5′ UTR unspliced in cell probed at different DMS concentrations. The data set was generated 20 times to simulate a read depth of 4,000 reads. Mutation count was performed (**a**) with indels and (**b**) without indels with increasing stringency on the per position quality score filter (-q). *Boxes represent quartile 1 (Q1) to quartile 3 (Q3). The second quartile (Q2) is marked by a line inside the box. Whiskers correspond to the box' edges +/−1.5 times the interquartile range (IQR: Q3-Q1).*

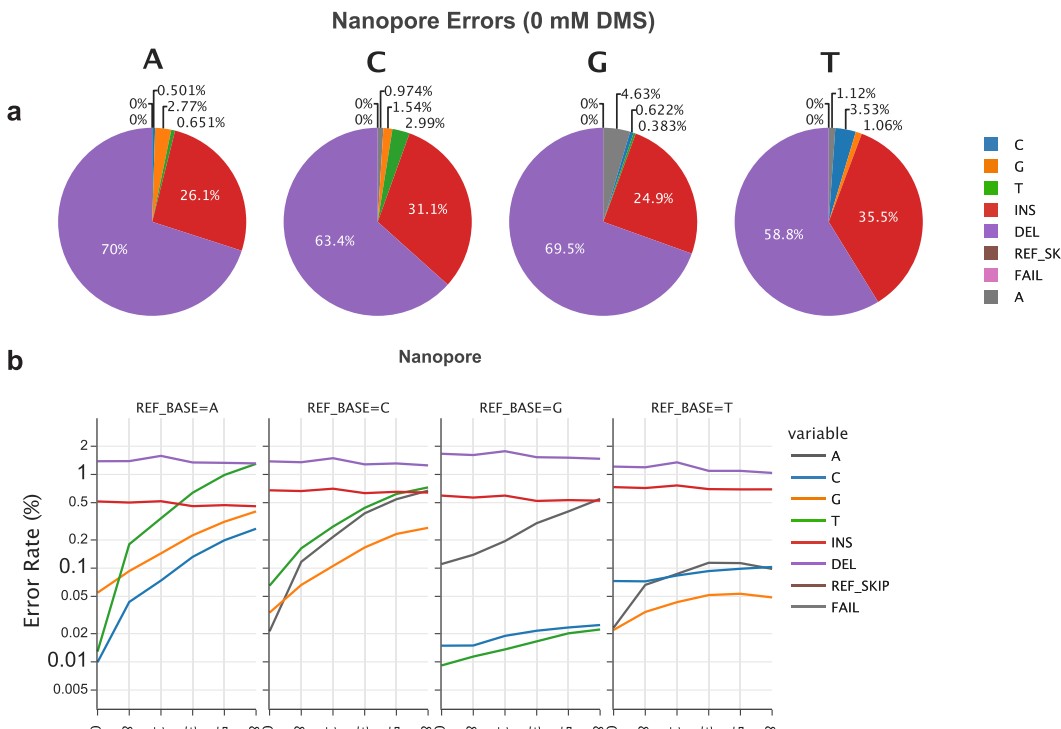

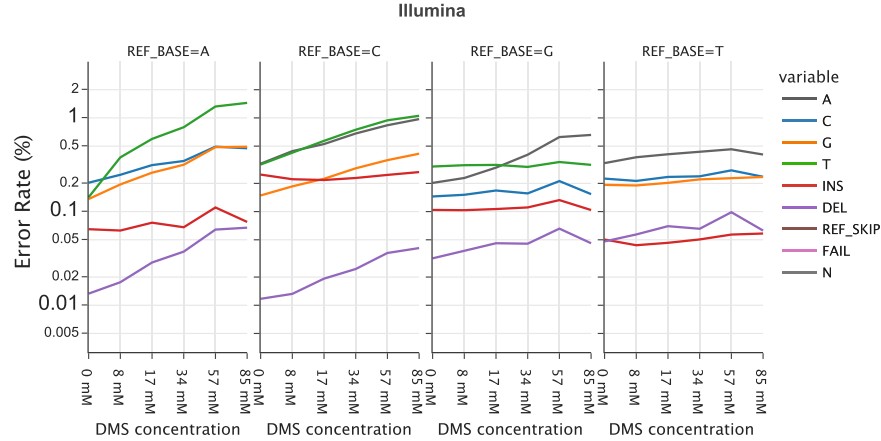

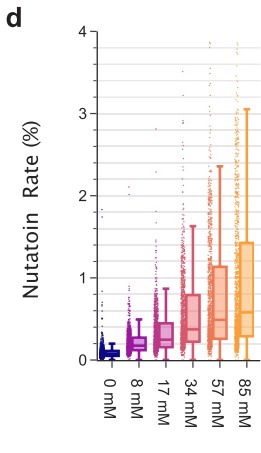

**Extended Data Fig. 4 | See next page for caption.**

**Extended Data Fig. 4 | Mutation type analysis of nanopore and Illumina data.** Unspliced HIV-1 RNA was probed in cells at different DMS concentrations in biologically independent replicates and matched Illumina and Nanopore sequencing experiments were performed. After analysis, errors were classified as mismatches (A, C, G, T), insertions (INS), deletions (DEL), or unclassified (REF_SKIP, FAIL, N). (**a**) Pie charts showing sequencing errors in the untreated nanopore sample. (**b-c**) Mutation type analysis for (**b**) nanopore and (**c**) Illumina data at A, C, G, and T nucleotides. Mean log10 error rates from alignments produced from (**b**) LAST or (**c**) bowtie (see Methods) are shown. (**d**) Box plot of mutation rates for each DMS concentration with optimized rfcount settings. Boxes represent quartile 1 (Q1) to quartile 3 (Q3). The second quartile (Q2) is marked by a line inside the box. Whiskers correspond to the box' edges +/−1.5 times the interquartile range (IQR: Q3-Q1). All data points are shown as dots.

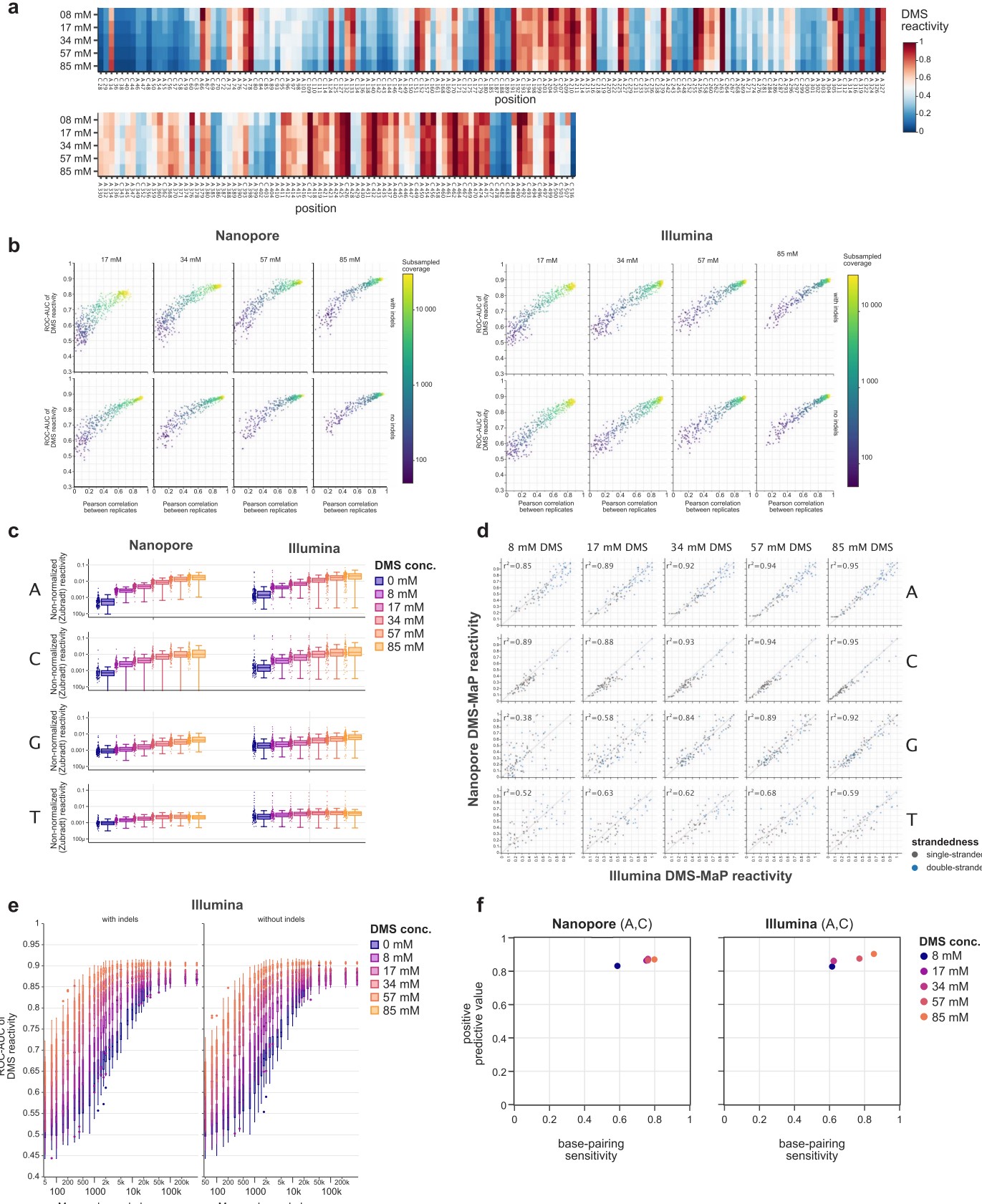

**Extended Data Fig. 5 | See next page for caption.**

**Extended Data Fig. 5 | DMS reactivities for the unspliced HIV-1 5'UTR and Gag coding sequence.** (**a**) A heatmap of mean DMS reactivities for the unspliced HIV-1 5'UTR at different DMS concentrations. DMS reactivities of A and C residues are shown using a blue-white-red color scheme. (**b**) Plots showing the relationship between ROC-AUC scores (y-axis) and the Pearson's correlation between both biological replicates (x-axis) of the US RNA at various DMS concentrations (columns). The analysis was performed with (top row) and without indels (bottom row). The color of each points represents the mean coverage obtained by sub-sampling. (**c**) Mutation rates after as detected with rfcount using per position filter of Q22 and ignoring indels for matched nanopore and Illumina data. Boxes represent quartile 1 (Q1) to quartile 3 (Q3). The second quartile (Q2) is marked by a line inside the box. Whiskers correspond to the box's edges +/−1.5 times the interquartile range (IQR: Q3-Q1). Boxes are colored according to the DMS concentration used in the experiment (**d**) Relationship between DMS reactivities for matched samples sequenced by Illumina (x-axis) and nanopore (y-axis) is shown for the US RNA in cells at various DMS concentrations and for each individual nucleotide. Pearson's correlation coefficient between datasets is also shown. Dots are colored to show strandedness of the nucleotide (gray double-, blue single-stranded) according to the reference model. (**e**) ROC-AUC scores of the match between Illumina DMS reactivities and the consensus structure for each DMS concentration as the mean coverage isis varied by sub-sampling (n=20). Data was analyzed with and without counting indels. Boxes represent quartile 1 (Q1) to quartile 3 (Q3). The second quartile (Q2) is marked by a line inside the box. Whiskers correspond to the box's edges +/−1.5 times the interquartile range (IQR: Q3-Q1). (**f**) Positive predictive value and base-pairing sensitivity of Nano-DMS-MaP and Illumina DMS-MaP reactivity (on A,C residues)used to guide Eternafold structure prediction of the first 380 nt of unspliced HIV 5' UTR in cell at increasing DMS concentrations. All DMS reactivities are mean of two biologically independent replicates.

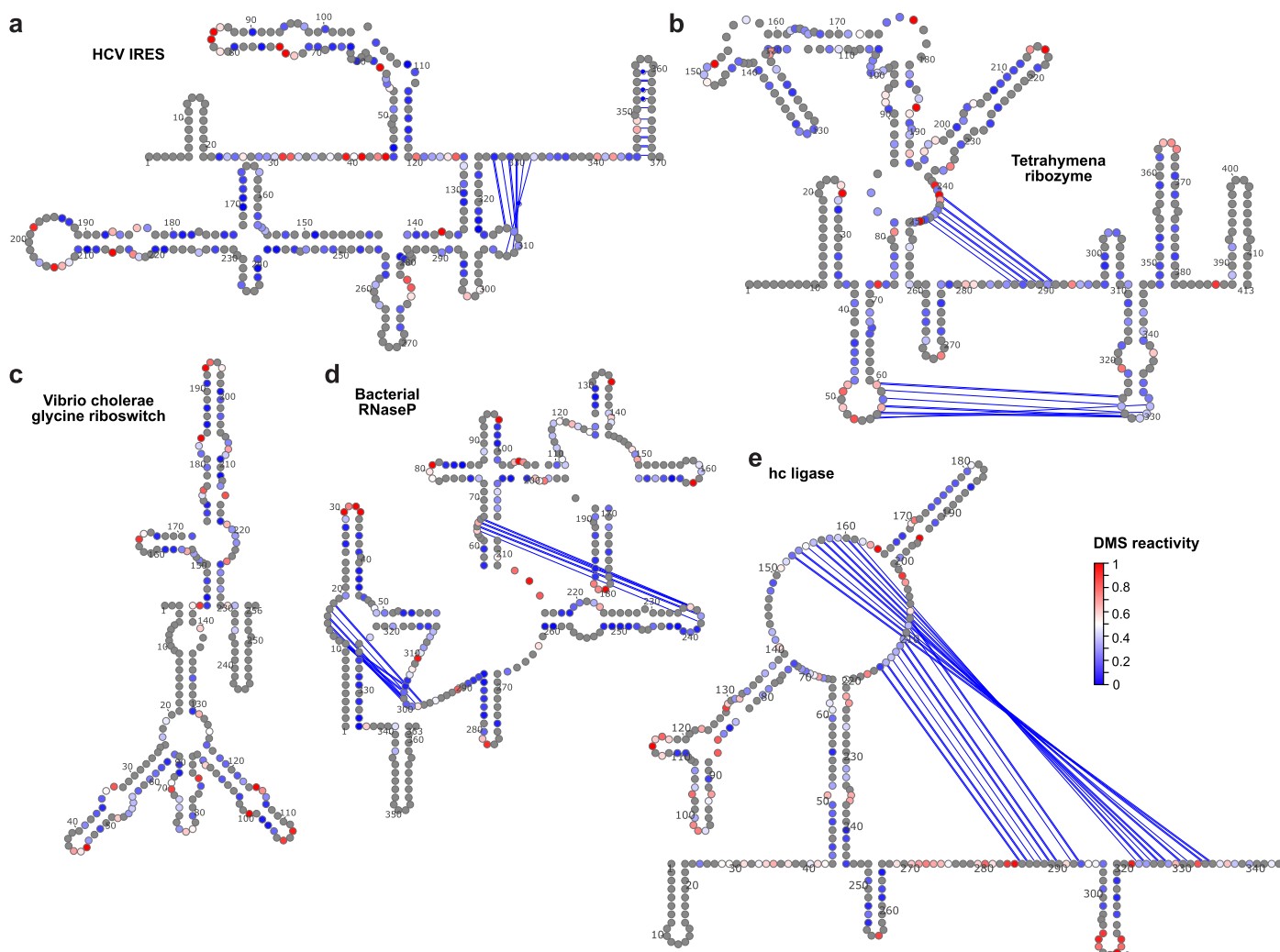

**Extended Data Fig. 6 | Benchmarking Nano-DMS-MaP on well-defined RNAs in vitro.** Benchmarking of Nano-DMS-MaP data was performed on 5 highly structure RNAs with well-defined secondary structures. DMS probing was 57 mM. (**a-e**) Secondary structures of the bact. RNaseP, hc16 ligase, HCV IRES, Tetrahymena Ribozyme and V. chol. Glycine riboswitch as reported. A and C residues are colored according to DMS reactivities using a blue-white-red color scheme. Blue is 0, white is 0.5, Red is 1. Red nucleotides are reactive to DMS indicating single strandedness. Blue nucleotides are unreactive to DMS indicating double-strandedness or protection from modification due to occlusion for example, by other protein or nucleic acid molecules. G and U are colored in gray and were not analyzed.

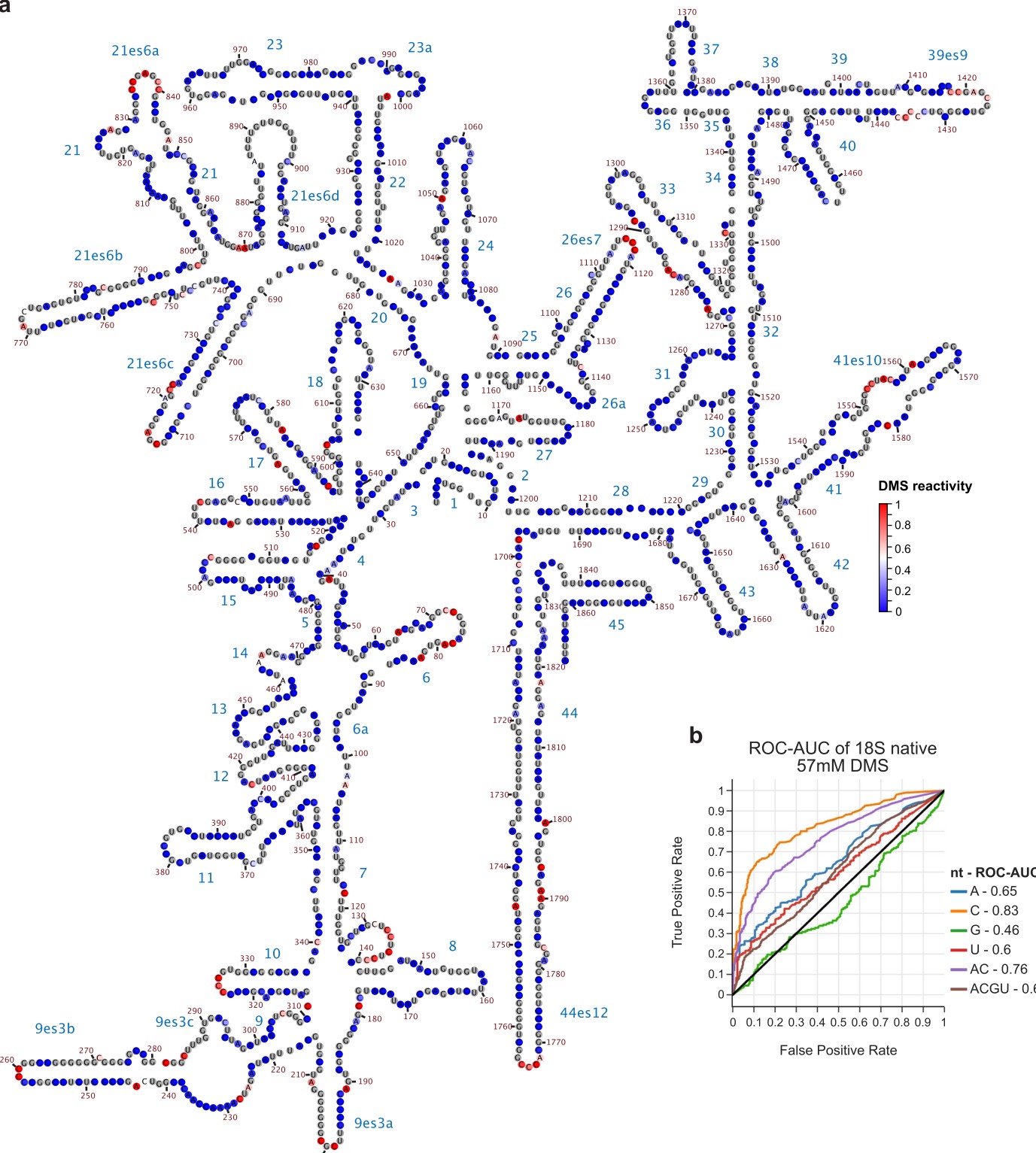

**Extended Data Fig. 7 | Benchmarking Nano-DMS-MaP on the human 18S ribosomal RNA in situ.** Secondary structure and ROC-AUC curves for the human 18 S ribosomal RNA probed in cell with 57 mM DMS. (**a**) The secondary structures of the 18 S RNA obtained from http://apollo.chemistry.gatech.edu/RiboVision2/ is colored according to DMS reactivities using a blue-white-red color scheme.

G and U are colored in gray and were not analyzed. (**b**) Receiver Operating Characteristic (ROC) curve representing the false/true positive rates of a binary classifier for strandedness as the DMS reactivity threshold is varied. ROC-AUC scores are shown for A, C, G, T residues or combinations thereof.

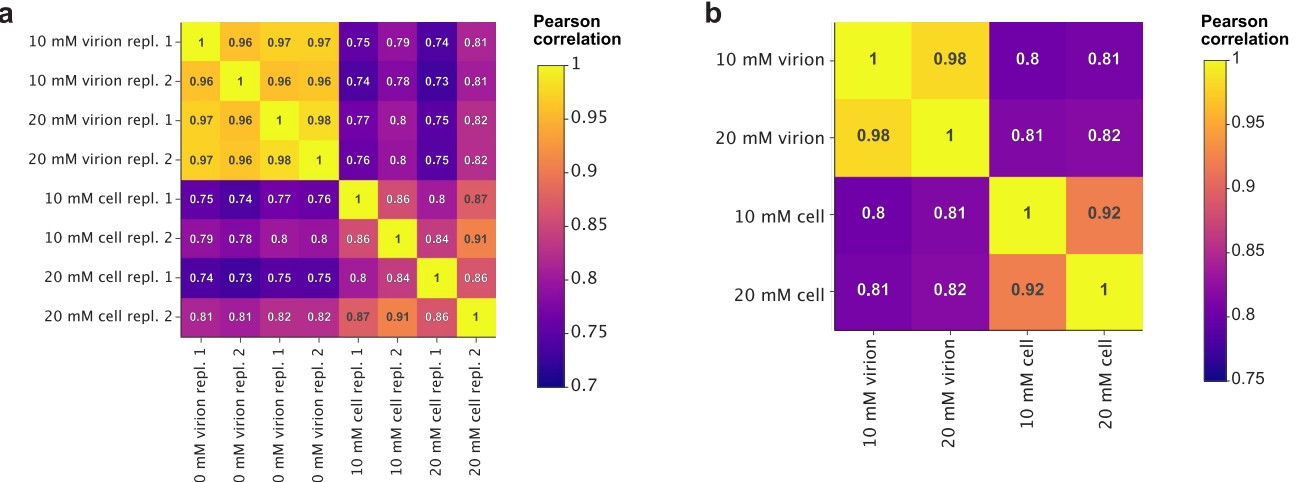

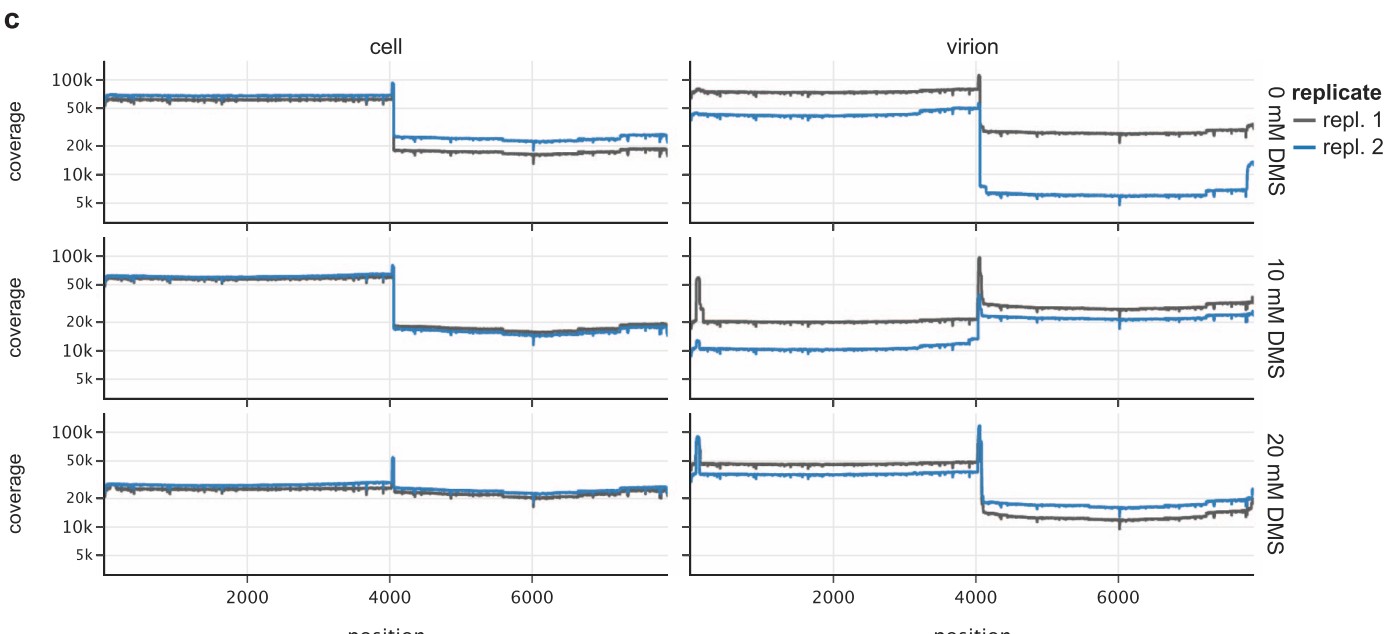

**Extended Data Fig. 8 | Near full length HIV-1 genome correlation and coverages.** (**a**) Heatmaps showing Pearson's correlation between DMS reactivities obtained for both biological replicates of in virion and cell RNA probed at 10 or 20 mM. (**b**) Pearson's correlation between mean reactivity of both biological replicates per sample. (**c**) Coverage plots for each virion and cell replicate at two DMS concentrations (10 mM and 20 mM) and the untreated sample (0 mM). Number of reads obtained in log scale (y-axis) per genome position (x-axis).

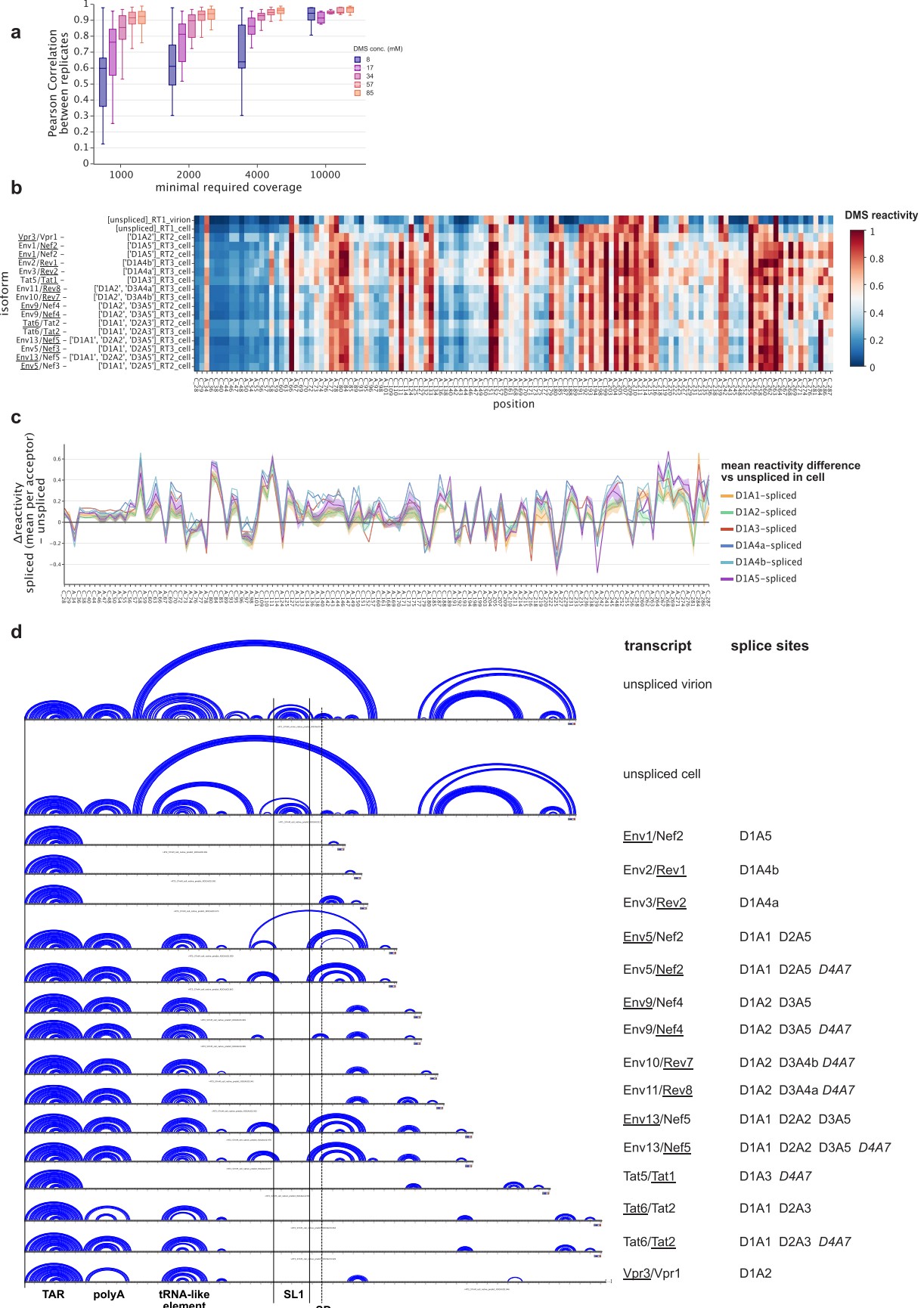

**Extended Data Fig. 9 | See next page for caption.**

**Extended Data Fig. 9 | Correlations, heat map analysis and structural predictions of spliced RNA.** (**a**) Correlation between DMS reactivities obtained from 2 independent experimental replicates of the unspliced HIV-1 RNA in cells. Minimal required read depth per isoform was set and correlation for each isoform passing the filter was evaluated. *Boxes represent quartile 1 (Q1) to quartile 3 (Q3). The second quartile (Q2) is marked by a line inside the box. Whiskers correspond to the box's edges +/−1.5 times the interquartile range (IQR: Q3-Q1).* (**b**) A heatmap of DMS reactivities after probing at 57 mM DMS concentration for the unspliced HIV-1 5'UTR in cells and virions, as well as individual reactivities for spliced transcripts in cells. *Reactivities of A and C residues are shown using a* *blue-white-red color scheme.* (**c**) *Line* plot showing ΔDMS reactivity (mean spliced RNA per first acceptor site – unspliced RNA) for 57 mM at each position in the RNA. Error bands indicate standard deviation within the spliced RNA with same first acceptor site. (**d**) Arc-plots representing *de novo* RNA structural predictions for individual spliced isoforms. Predictions were performed using EternaFold guided by DMS reactivities. Underscore labels the actual isoform, depending on whether RNA had been reverse transcribed with RT-PS or RT-FS. For Vpr3 only the structure of the first 564 nt is shown. Varna files are provided in Supplementary Data Set 2.

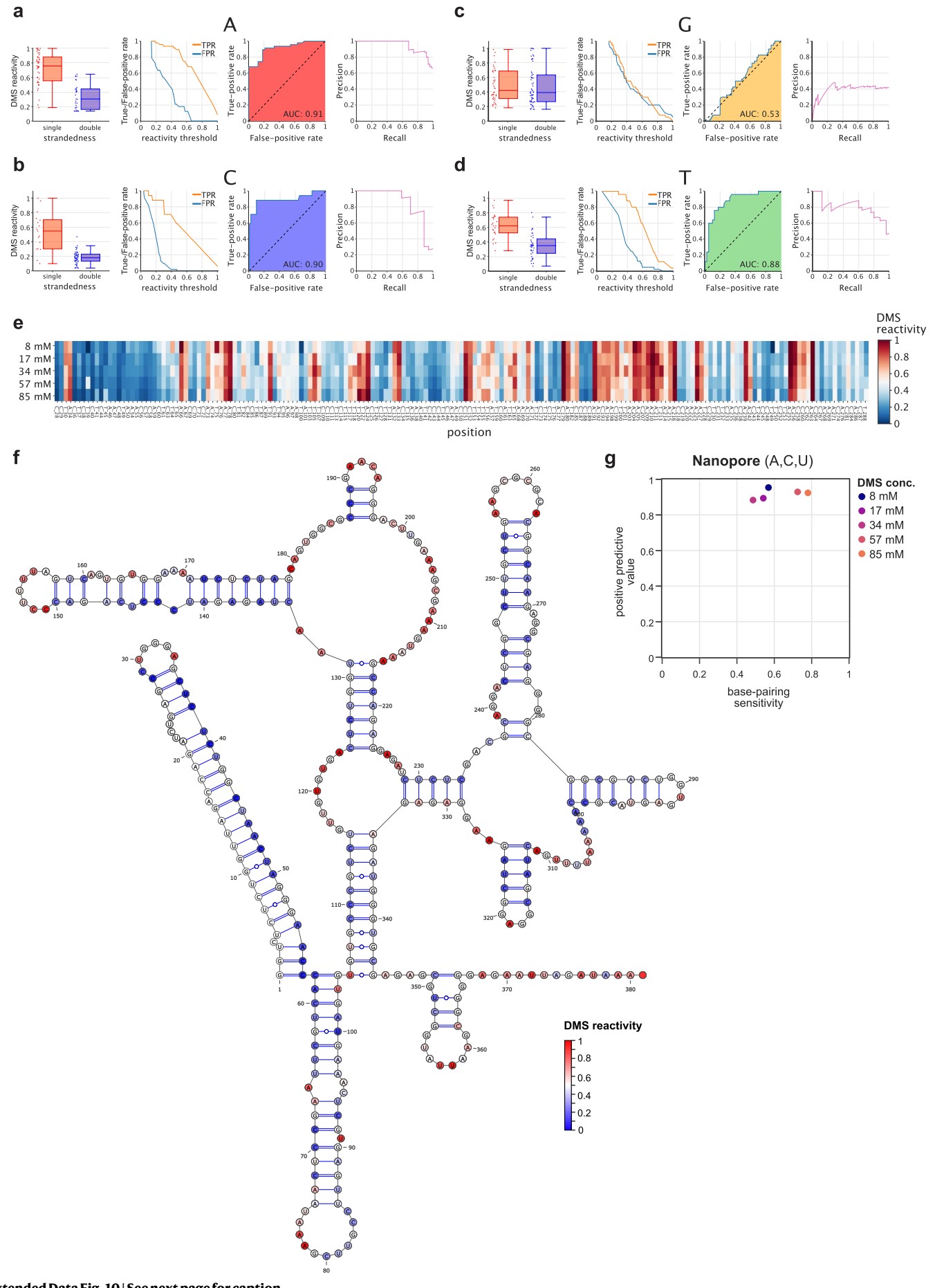

**Extended Data Fig. 10 | See next page for caption.**

**Extended Data Fig. 10 | RNA structural information at U residues.** Box plots showing DMS reactivity distributions for single stranded and double stranded RNA, false positive and true positive against threshold, receiver operator characteristic curves and precision against recall curves for one of the two independent replicates of unspliced HIV RNA from cells treated with 85 mM DMS for (**a**) A,(**b**), C (**c**), G, and (**d**) U residues. (**e**) Heat maps of DMS reactivities at A, C, and U residues for the unspliced HIV-1 RNA in cells at increasing DMS concentration for one of the biologically independent replicates. (**f**) DMS reactivities of unspliced HIV RNA from cells treated with 85 mM DMS at A, C, U

residues plotted onto the consensus structure of the HIV-1 5'UTR in the unspliced RNA. A, C and U residues are colored according to DMS reactivities using a blue-white-red color scheme. Boxes represent quartile 1 (Q1) to quartile 3 (Q3). The second quartile (Q2) is marked by a line inside the box. Whiskers correspond to the box's edges +/−1.5 times the interquartile range (IQR: Q3-Q1). (**g**) Positive predictive value and base-pairing sensitivity of reactivity-guided Eternafold structure prediction of the first 380 nt of the unspliced HIV 5' UTR using mean in cell Nano-DMS-MaP reactivity at A, C and U residues of both biologically independent replicates for increasing DMS concentrations.

# nature research

# Reporting Summary

Nature Research wishes to improve the reproducibility of the work that we publish. This form provides structure for consistency and transparency in reporting. For further information on Nature Research policies, see our Editorial Policies and the Editorial Policy Checklist.

## Statistics

For all statistical analyses, confirm that the following items are present in the figure legend, table legend, main text, or Methods section.

| n/a | Confirmed | |
|---|---|---|
| ☐ | ☒ | The exact sample size (*n*) for each experimental group/condition, given as a discrete number and unit of measurement |
| ☐ | ☒ | A statement on whether measurements were taken from distinct samples or whether the same sample was measured repeatedly |
| ☒ | ☐ | The statistical test(s) used AND whether they are one- or two-sided *Only common tests should be described solely by name; describe more complex techniques in the Methods section.* |
| ☒ | ☐ | A description of all covariates tested |
| ☒ | ☐ | A description of any assumptions or corrections, such as tests of normality and adjustment for multiple comparisons |
| ☐ | ☒ | A full description of the statistical parameters including central tendency (e.g. means) or other basic estimates (e.g. regression coefficient) AND variation (e.g. standard deviation) or associated estimates of uncertainty (e.g. confidence intervals) |
| ☒ | ☐ | For null hypothesis testing, the test statistic (e.g. *F*, *t*, *r*) with confidence intervals, effect sizes, degrees of freedom and *P* value noted *Give P values as exact values whenever suitable.* |
| ☒ | ☐ | For Bayesian analysis, information on the choice of priors and Markov chain Monte Carlo settings |
| ☒ | ☐ | For hierarchical and complex designs, identification of the appropriate level for tests and full reporting of outcomes |
| ☐ | ☒ | Estimates of effect sizes (e.g. Cohen's *d*, Pearson's *r*), indicating how they were calculated |

*Our web collection on statistics for biologists contains articles on many of the points above.*

## Software and code

Policy information about availability of computer code

| | |
|---|---|
| Data collection | Data was generated on Oxford Nanopore Technologies Minion Mk1B using Minknow software version (21.11.8) and a R10.4 MinIon flow cell (FLO-MIN112). |
| Data analysis | Data was basecalled with guppy 6.1.3. Virtual gels were generated with a custom python v3.8.5 script using the numpy library version 1.19.2. Read to isoform mapping was performed using IsoQuant version 2.0. Alignments were performed on specific reference sequences using LAST version 1419. sam files were then processed using samtools version 1.12. Mismatch patterns were analyzed from BAM alignment files with perbase version 0.8.3 and custom python scripts. Mutational Profiling analysis was performed for each isoform separately using RNAFramework version 2.7.2 or custom python scripts. Correlation scores of reactivity profiles was calculated using the python package scikit-learn version 0.21.3. Sub-figures were generated using the python library plotly version 4.14.3. All custom scripts will be supplied before publication on GitHub (https://github.com/smyth-lab/Nano-DMS-MaP). De novo RNA structure prediction was performed with Eternafold version 1.3.1. RNA structures were visualised using VARNA version 3.93. |

For manuscripts utilizing custom algorithms or software that are central to the research but not yet described in published literature, software must be made available to editors and reviewers. We strongly encourage code deposition in a community repository (e.g. GitHub). See the Nature Research guidelines for submitting code & software for further information.

## Data

Policy information about availability of data

All manuscripts must include a data availability statement. This statement should provide the following information, where applicable:

- Accession codes, unique identifiers, or web links for publicly available datasets
- A list of figures that have associated raw data
- A description of any restrictions on data availability

All basecalled data are available on the sequence read archive (SRA) at Bioproject accession number PRJNA938445 and Sequencing Project Number SRP424422

# Field-specific reporting

Please select the one below that is the best fit for your research. If you are not sure, read the appropriate sections before making your selection.

☒ Life sciences   ☐ Behavioural & social sciences   ☐ Ecological, evolutionary & environmental sciences

For a reference copy of the document with all sections, see nature.com/documents/nr-reporting-summary-flat.pdf

# Life sciences study design

All studies must disclose on these points even when the disclosure is negative.

| | |
|---|---|
| Sample size | We performed two biologically independent experimental replicates. This sample size was chosen because of previously known high correlation between biological replicates. |
| Data exclusions | All reads with a mean read quality score below 10 and those not demultiplexed by the basecaller were removed, as described in the manuscript. |
| Replication | We assessed reproducibility by calculating Pearson's correlation coefficient between two experimental replicates for relative abundance of transcript isoforms and for DMS reactivities. All replications were successful, and indicated in the manuscript. |
| Randomization | Randomisation was not performed as it was not relevant to our study. |
| Blinding | Blinding was not performed as it was not relevant to our study. |

# Reporting for specific materials, systems and methods

We require information from authors about some types of materials, experimental systems and methods used in many studies. Here, indicate whether each material, system or method listed is relevant to your study. If you are not sure if a list item applies to your research, read the appropriate section before selecting a response.

## Materials & experimental systems

| n/a | Involved in the study |
|---|---|
| ☒ | ☐ Antibodies |
| ☐ | ☒ Eukaryotic cell lines |
| ☒ | ☐ Palaeontology and archaeology |
| ☒ | ☐ Animals and other organisms |
| ☒ | ☐ Human research participants |
| ☒ | ☐ Clinical data |
| ☒ | ☐ Dual use research of concern |

## Methods

| n/a | Involved in the study |
|---|---|
| ☒ | ☐ ChIP-seq |
| ☒ | ☐ Flow cytometry |
| ☒ | ☐ MRI-based neuroimaging |

## Eukaryotic cell lines

Policy information about cell lines

| | |
|---|---|
| Cell line source(s) | HEK 293T cells were obtained from the Caliskan laboratory. |
| Authentication | HEK 293T cells were not independently authenticated. |
| Mycoplasma contamination | HEK 293T cells are tested for mycoplasma infection monthly, and were consistently negative. |
| Commonly misidentified lines (See ICLAC register) | N/A |

