## [Peer Review File · Nature Methods]

Peer Review Information

Manuscript Title: Nano-DMS-MaP-seq allows isoform specific RNA structure determination

Corresponding author name(s): Redmond Smyth

Editorial Notes: n/a

Reviewer Comments & Decisions:

Decision Letter, initial version:

Dear Dr Smyth,

Please accept our sincere apologies for the length of time that it has taken to get the decision on your manuscript. Your Article entitled "Nano-DMS-MaP-seq allows isoform specific RNA structure determination" has now been seen by 3 reviewers, whose comments are attached. In the light of their advice we have decided that we cannot offer to publish your manuscript in Nature Methods.

You will see that, while they find your work of some potential interest, the reviewers raise concerns about the advance Nano-DMS-MaP-seq represents over available methods and about its broad applicability. We think that these criticisms, particularly the limited length of the RNA probed, are sufficiently important as to prevent publication of your work in Nature Methods.

I am sorry that we cannot be more positive on this occasion but hope that you find the reviewers' comments helpful when preparing your paper for submission elsewhere.

Sincerely,
Lei

Lei Tang, Ph.D.
Senior Editor
Nature Methods

Reviewer Comments:

Reviewer #1 (Remarks to the Author):

The manuscript by Bohn et al describes a new RNA chemical probing strategy read-out by nanopore sequencing, which they name Nano-DMS-MaP-seq. As an initial application, they apply Nano-DMS-MaP-Seq to define isoform-specific structural variation in HIV-1. The most significant aspect of this manuscript is the finding that nanopore-sequencing error rates, which are normally very high, can be greatly reduced through a relatively simple signal filtering scheme. This is an important insight that will likely have broad utility for a diverse genomics experiments. Nevertheless, the authors fail to convincingly show that this advance enables new understanding or discoveries, as is expected of Nature Methods papers. They only characterize a single RNA system (HIV-1) and their benchmarking does not meet the standard needed to make Nano-DMS-MaP-seq a convincing, general-purpose replacement for DMS-MaP. Further, while the authors correctly note the advantages offered by nanopore sequencing for measuring long RNAs, almost all of their data comes from amplicons <600 bp in length. Thus, their data very likely could have been obtained using existing Illumina platforms. The authors use Nano-DMS-MaP-seq to define structural changes in the HIV 5'UTR associated with splicing, but their attempts at structural modeling indicate that these DMS reactivity changes do not entail a change in global structure. While this may reflect limitations of structural modeling algorithms, as it stands these data do not significantly advance understanding of mechanism. It is also not clear whether these observations are novel. Michael Summers and Alice Telesnitsky have proposed detailed models for how 5'UTR structure changes drive packaging (PMID: 32327595); the authors fail to discuss how their observations fit within this model. In summary, while the advance in nanopore sequencing is exciting, major additional experiments and analyses are needed for the manuscript to meet Nature Methods significance standards.

Major critiques

-The authors only apply Nano-DMS-MaP-Seq to a single RNA system. Additionally, the SHAPE-based HIV-1 RNA structure the authors use as a reference is not technically "known" and is disputed by other groups. For example, see multiple papers from Michael Summers' lab. At minimum, the authors should validate Nano-DMS-MaP on 1-2 additional RNAs with known structures, such as the E. coli rRNA, or a large riboswitch or ribozyme system.

-The ability to use DMS data to accurately reconstruct RNA structure is critical to the utility of these methods. The authors briefly discuss that they used their data for structural modeling, but this data is buried in supplemental datasets and not properly quantified. These data need to be clearly described in the main text using standard modeling statistics such as base pairing PPV and sensitivity. As noted above, the authors should validate the accuracy of their de novo structural modeling using tests on multiple accepted reference RNAs.

-With their filtering scheme, the authors achieve an error rate of ~0.1% in the ethanol-treated sample. It would be good to know what fraction of this error is due to sequencing error versus error introduced by MarathonRT under Mn²⁺ conditions. The authors could use the same RT-PCR products, but then sequence them using Illumina sequencing for a direct comparison. Similarly, it would be good to know how the measured DMS reactivities differ from those measured by Illumina platforms.

-The authors need to discuss their results on HIV-1 structural heterogeneity in relation to the models proposed by Michael Summers and Alice Telesnitsky. For example, see PMID: 32647061 or PMID: 32327595

Minor critiques

-The authors name their method Nano-DMS-MaP-Seq. This name is unwieldy and will further the unnecessary proliferation of the redundant “seq” suffix. Sequencing is implied by “MaP”. The paper by Guo et al on MarathonRT calls their method DMS-MaP. The Weeks lab also calls their DMS protocol DMS-MaP. SHAPE experiments use the name SHAPE-MaP. Only TGIRT-based DMS protocols use the -seq suffix. Since the authors are not using the TGIRT enzyme, I encourage them to name their method Nano-DMS-MaP to maintain consistency with the broader field.

-It is impossible to read many of the supplementary figures because they are at too low of a resolution.

-In their discussion of correlated chemical probing, the authors should also cite key work on base-pair detection from the Weeks lab such as PMID: 31744869 or PMID: 35320755.

-The schematic in 4b is confusing – the authors should clearly show which sequences are variable based on isoform. For example, is the pseudoknot able to form in all isoforms?

-The use of “error rate” in Figure 2 is confusing, because error is typically associated with an undesirable outcome. “Mutation rate” or “mismatch rate” would be a less confusing terminology.

-Figure 2D is challenging to interpret, as both G/U (no data) and A/C nts with reactivities = 0.5 are shown in white. It would be easier to interpret if the authors showed the sequence and used a different color scheme to denote G/U.

-Figure 2E and 2F seem redundant with the ROC analysis shown in Figure 2G. Since the authors are not using hard reactivity cutoffs for calling base-paired versus single stranded, and indeed using such cutoffs generally harms data interpretation, panels 2E and 2F should be dropped.

-The authors interchangeably use partially spliced (PS) and singly spliced (SS). This is quite confusing. They should choose one name and use it uniformly.

-The acronym SFB in methods is not defined.

-The methodology used for structural modeling is poorly described. What is the underlying folding algorithm of rf-fold? Did the authors make sure to only include A/C nucleotides? How were G/U nts masked out? What method did they use to convert DMS reactivities into energetic restraints for folding?

Typos

The caption to Figure 1 refers to (a) and (b) panels, but the figure does not have a (b) label.

There is a typo in the Figure S3 caption: panel (a) is listed twice?

Reviewer #2 (Remarks to the Author):

The authors develop Nanopore based DMS-MaPseq to detect isoform specific RNA structure. The benefits of Nanopore sequencing are that it works on very long molecules and it can directly sequence RNA and in principle detect the DMS methylation without any conversion. The challenges are that nanopore sequencing is low accuracy relative to Illumina platforms, the base-callers require training, and it is expensive per read. The authors use some middle-ground where they sequence cDNA molecules. They troubleshoot conditions and find that Marathon +Manganese can read through the DMS modifications and produce long cDNA molecules. The authors do a lot of increase the signal over noise e.g. quality filtering, optimizing read depth, and taking only mismatches as signal. They demonstrate good agreement with known HIV structures and their data suggests structural implications on splicing. Overall, the work is novel and the approach is of vast interest. However, the authors must address some key questions before I would recommend publication:

1. How well does it work compared to highly accurate sequencing (e.g. Illumina platform)? It would be great if the authors can take the exact same cDNA library, fragment it and sequence it by established protocols to show that there is no/minimal bias and signal loss in their new sequencing platform.
2. The use of manganese is concerning because it makes the RT reaction error prone. The authors should compare an untreated control with and without manganese to assess how much noise comes from use of manganese.
3. It would be great if the authors can compare Marathon +Mn with the one that has been published previously using TGIRT enzyme (Zubradt et al), which avoids Manganese.

4. Since HIV-1 has been probed with DMS in virion and in cells (Tomezsko et al 2020) the authors should directly compare DMS signal over the 5'UTR and if available over the whole genome inside the virion (where there is just the full length molecule) with published results. In principle the two strategies should produce the exact same DMS signal and if there are discrepancies it would be important to know the source.

5. Somethings that were not clear to me – what is the longest piece that was sequenced with high quality data, can the authors get the entire HIV-1 genome on one molecule? It seemed that the majority of the data is coming from 500-1kb, please explain. Also a histogram of number of mutations per molecule would be very useful.

I suggest that you consider Nature Communications as a suitable venue for your work. To transfer your manuscript there, please use our <https://mts-nmeth.nature.com/cgi-bin/main.plex?el=A1M3Vvr3A3ysy7X5A9ftdPb9mVnNbjZJlgCZTKPQgZ> manuscript transfer portal. You will not have to re-supply manuscript metadata and files, unless you wish to make modifications, but please note that this link can only be used once and remains active until used. For more information, please see our http://www.nature.com/authors/author_resources/transfer_manuscripts.html?WT.mc_id=EMI_NPG_1511_AUTHORTRANSF&WT.ec_id=AUTHOR manuscript transfer FAQ page.

Note that any decision to opt in to In Review at the original journal is not sent to the receiving journal on transfer. You can opt in to *[In Review](https://www.nature.com/nature-portfolio/for-authors/in-review)* at receiving journals that support this service by choosing to modify your manuscript on transfer. In Review is available for primary research manuscript types only.

** For Nature Portfolio general information and news for authors, see <http://npg.nature.com/authors>.

Springer Nature Author Services can help you develop and improve your manuscript for your next submission through services including **English language editing, developmental comments, manuscript formatting, figure preparation, translation**, and more. https://authorservices.springernature.com/go/sn/?utm_source=EJP&utm_medium=Rejection+Email&utm_campaign=SNAS+Referrals+2022&utm_id=ref2022 Find out more about how you can take advantage of these services.

Please note that the use of these tools, or any other service, is not a requirement for publication, nor does it imply or guarantee that editors will accept the article, or even select it for peer review.

Author Rebuttal to Initial comments

Reviewer Comments:

Reviewer #1 (Remarks to the Author):

The manuscript by Bohn et al describes a new RNA chemical probing strategy read-out by nanopore sequencing, which they name Nano-DMS-MaP-seq. As an initial application, they apply Nano-DMS-MaP-Seq to define isoform-specific structural variation in HIV-1. The most significant aspect of this manuscript is the finding that nanopore-sequencing error rates, which are normally very high, can be greatly reduced through a relatively simple signal filtering scheme. This is an important insight that will likely have broad utility for a diverse genomics experiments. Nevertheless, the authors fail to convincingly show that this advance enables new understanding or discoveries, as is expected of Nature Methods papers.

They only characterize a single RNA system (HIV-1) and their benchmarking does not meet the standard needed to make Nano-DMS-MaP-seq a convincing, general-purpose replacement for DMS-MaP.

We now perform benchmarking of our method on 5 highly structured RNAs with well described secondary structures (HCV IRES; Tetrahymena ribozyme, OncomiR-1, Bact. Rnase P, hc16 ligase). The results of these experiments confirm that Nano-DMS-MaP recovers high quality structural information on a wide variety of RNAs. Additionally, we will provide a direct comparison with DMS-MaP data from Illumina sequencing to show equivalency in data quality.

Further, while the authors correctly note the advantages offered by nanopore sequencing for measuring long RNAs, almost all of their data comes from amplicons <600 bp in length. Thus, their data very likely could have been obtained using existing Illumina platforms.

We now provide additional data to show that Nano-DMS-MaP can probe long RNAs. Namely, two amplicons of the HIV-1 genomic RNA of approximately 4.5kb, and the human 18S rRNA of 1.9kb.

We would like to stress again that in contrast to DMS-MaP, which relies on Illumina sequencing, Nano-DMS-MaP achieves long read chemical probing needed for isoform resolved structural analysis. To differentiate between all HIV-1 mRNA isoforms each sequencing read must cover all potential splice sites, in addition to the common sequence of the 5' UTR. Thus, while we could have mapped Illumina sequencing data to some of the shorter isoforms, isoform resolved RNA structural analysis of the HIV splicing landscape would not be feasible with Illumina data only. This is also depicted in Figure 1a of our manuscript.

The authors use Nano-DMS-MaP-seq to define structural changes in the HIV 5'UTR associated with splicing, but their attempts at structural modeling indicate that these DMS reactivity changes do not entail a change in global structure. While this may reflect limitations of structural modeling algorithms, as it stands these data do not significantly advance understanding of mechanism. It is also not clear whether these observations are novel. Michael Summers and Alice Telenitsky have proposed detailed models for how 5'UTR structure changes drive packaging (PMID: 32327595); the authors fail to discuss how their observations fit within this model. In summary, while the advance in nanopore sequencing is exciting, major additional experiments and analyses are needed for the manuscript to meet Nature Methods significance standards.

Michael Summers and Alice Telenitsky have contributed important structural models of the HIV-1 *genomic* RNA, showing how 5'UTR structural changes influence genome packaging. However, these models provide no information on the structure of the 5'UTR within HIV-1 *spliced* viral RNAs, and it remains unclear how

these spliced transcripts avoid being accidentally packaged into viral particles. We hypothesized that spliced transcripts have different structures within the common 5'UTR region compared to genomic RNA that might explain why spliced viral RNAs are excluded from being packaged into viral particles

To test this hypothesis, we performed isoform resolved structural analysis of over 20 spliced viral transcripts. We show that *all* spliced transcripts have nucleotides within the stem of the major packaging structure (at SL1) that are reactive to DMS. This DMS reactivity data proves that stem loop is restructured in spliced viral RNAs compared to the genomic RNA. This is a novel, and substantial discovery which will be discussed in the context of the Summer's and Telenitsky models in the revised manuscript.

Major critiques

-The authors only apply Nano-DMS-MaP-Seq to a single RNA system. Additionally, the SHAPE-based HIV-1 RNA structure the authors use as a reference is not technically "known" and is disputed by other groups. For example, see multiple papers from Michael Summers' lab. At minimum, the authors should validate Nano-DMS-MaP on 1-2 additional RNAs with known structures, such as the E. coli rRNA, or a large riboswitch or ribozyme system.

In our original manuscript, we validated the Nano-DMS-MaP on a 'consensus' structure of the HIV-1 genome from Kevin Weeks. We acknowledge that alternative structures have been proposed and therefore agree with the reviewer that it is important to validate Nano-DMS-MaP further.

We have now done so using 5 structured RNAs (HCV IRES; Tetrahymena ribozyme, OncomiR-1, Bact. RNase P, hc16 ligase) *in vitro* (Figure R1) and the human ribosomal 18S rRNA in cells (Figure R2). The structures of these RNAs have previously been solved by crystallography or cryoEM. Our results confirm that Nano-DMS-MaP recovers high quality structural information on a wide variety of RNAs of different lengths, both *in vitro* and on native structures in cells.

Figure R1: Preliminary Nano-DMS-MaP analysis of five structured RNAs. (a) ROC-AUC scores measure the match between a known secondary structure and the DMS reactivities obtained in Nano-DMS-MaP experiments. ROC-AUC of approximately 0.8 [0.89-0.74] are obtained indicating good match between DMS and structure. (b) Visual representation of DMS reactivities on the known Bact. RNase P structure. Red indicates high DMS reactivity and single strandedness. Blue indicated low DMS reactivity and double strandedness.

Figure R2: Preliminary Nano-DMS-MaP analysis of the 1.9kb 18S RNA/protein complex probed natively in cells. Visual representation of DMS reactivities on the known 18S structure. Red indicates high DMS reactivity and single strandedness. Blue indicated low DMS reactivity and double strandedness. ROC-AUC scores of 0.74 are obtained for A and C residues.

-The ability to use DMS data to accurately reconstruct RNA structure is critical to the utility of these methods. The authors briefly discuss that they used their data for structural modeling, but this data is buried in supplemental datasets and not properly quantified. These data need to be clearly described in the main text using standard modeling statistics such as base pairing PPV and sensitivity. As noted above, the authors should validate the accuracy of their *de novo* structural modeling using tests on multiple accepted reference RNAs.

In our original manuscript, we were able to *de novo* recover one of the 'consensus' models of the HIV-1 5'UTR, as determined by chemical probing (principally SHAPE). In the revised manuscript, we show that Nano-DMS-MaP has equivalent performance to DMS-MaP-seq on a range of highly structured RNAs, and that there is very good correlation between Nano-DMS-MaP reactivities and base-pairing in these reference RNAs.

Whilst we agree that accurate structural models of the spliced RNAs contribute to mechanistic understandings, one-dimensional chemical probing data is not always sufficient to determine high accuracy 2/3D RNA structures, regardless of the read-out method. Indeed, the HIV-1 field is replete with structural models of the 5'UTR based on chemical probing data (highlighted in a review by Summers PMID: 21762803). Many of these models are unsupported by orthogonal experiments and are most likely incorrect. We feel it important to avoid overinterpretation of the Nano-DMS-MaP data with regards to *de novo* structural modelling, and instead plan to generate high confidence structural models of spliced RNAs

in follow up work e.g. by using correlated chemical probing or the computational deconvolution of structural ensembles. Nevertheless, structural probing data generally enables measurement of RNA structure dynamics within cells in their native context, something that other methods lack.

With their filtering scheme, the authors achieve an error rate of ~0.1% in the ethanol-treated sample. It would be good to know what fraction of this error is due to sequencing error versus error introduced by MarathonRT under Mn²⁺ conditions. The authors could use the same RT-PCR products, but then sequence them using Illumina sequencing for a direct comparison. Similarly, it would be good to know how the measured DMS reactivities differ from those measured by Illumina platforms.

This is an interesting experiment that was also suggested by reviewer 2. We are currently performing Illumina sequencing of RT-PCR products from the unspliced 5' UTR and the 5 structured RNAs we used for benchmarking to assess the match between DMS-MaP and Nano-DMS-MaP (awaiting sequencing results).

-The authors need to discuss their results on HIV-1 structural heterogeneity in relation to the models proposed by Michael Summers and Alice Telesnitsky. For example, see PMID: 32647061 or PMID: 32327595

The cited manuscripts do not provide structural models of the HIV-1 5'UTR in spliced RNAs, but of HIV genomic RNA. Nevertheless, in the revised manuscript we will provide better context to our work by discussing our results with regards to the Summers and Telenitsky models, as well as other proposed structural models.

Minor critiques:

-The authors name their method Nano-DMS-MaP-Seq. This name is unwieldy and will further the unnecessary proliferation of the redundant "seq" suffix. Sequencing is implied by "MaP". The paper by Guo et al on MarathonRT calls their method DMS-MaP. The Weeks lab also calls their DMS protocol DMS-MaP. SHAPE experiments use the name SHAPE-MaP. Only TGIRT-based DMS protocols use the -seq suffix. Since the authors are not using the TGIRT enzyme, I encourage them to name their method Nano-DMS-MaP to maintain consistency with the broader field.

We agree that Nano-DMS-MaP is an appropriate name.

-It is impossible to read many of the supplementary figures because they are at too low of a resolution.

We apologize for the low-quality figures. We will ensure the supplementary figures are of higher quality in the revised manuscript.

-In their discussion of correlated chemical probing, the authors should also cite key work on base-pair detection from the Weeks lab such as PMID: 31744869 or PMID: 35320755.

These publications will be cited in the revised manuscript.

-The schematic in 4b is confusing – the authors should clearly show which sequences are variable based on isoform. For example, is the pseudoknot able to form in all isoforms?

The schematic in 4b depicts only the genomic RNA. Actually, the pseudoknot is absent in all isoforms. The figure legend of this schematic will be improved to clarify this point.

-The use of "error rate" in Figure 2 is confusing, because error is typically associated with an undesirable outcome. "Mutation rate" or "mismatch rate" would be a less confusing terminology.

The figures will be updated to 'mutation rate'.

-Figure 2D is challenging to interpret, as both G/U (no data) and A/C nts with reactivities = 0.5 are shown in white. It would be easier to interpret if the authors showed the sequence and used a different color scheme to denote G/U.

We will provide an updated figure with G/U residues in grey. A corresponding figure that includes the sequence was already provided in the original manuscript as a supplementary figure, because we believed that the additional sequence information makes the figure more difficult to read.

-Figure 2E and 2F seem redundant with the ROC analysis shown in Figure 2G. Since the authors are not using hard reactivity cutoffs for calling base-paired versus single stranded, and indeed using such cutoffs generally harms data interpretation, panels 2E and 2F should be dropped.

We agree with the reviewer that panel F provides redundant information. Panel E graphically shows that the best classification of the data into single stranded and double stranded RNA occurs using a threshold of 0.5, which provides rationale for the colour scheme used in Fig 2d.

-The authors interchangeably use partially spliced (PS) and singly spliced (SS). This is quite confusing. They should choose one name and use it uniformly.

We will carefully go through the manuscript to ensure only partially spliced (PS) is used.

-The acronym SFB in methods is not defined.

The acronym short fragment buffer (SFB) will be clarified on first use in the revised manuscript.

-The methodology used for structural modeling is poorly described. What is the underlying folding algorithm of rf-fold? Did the authors make sure to only include A/C nucleotides? How were G/U nts masked out? What method did they use to convert DMS reactivities into energetic restraints for folding?

We used rf-fold, which is a peer-reviewed and open-source software for facilitating RNA structure prediction from chemical probing data sets. The structural predictions provided in the original manuscript used default parameters and are therefore based on ViennaRNA using information only at A and C nucleotides.

The details regarding the folding algorithm, G/U masking, and conversion of DMS reactivities will be expanded upon in the revised manuscript.

Typos

The caption to Figure 1 refers to (a) and (b) panels, but the figure does not have a (b) label.

This will be corrected in the revised manuscript.

There is a typo in the Figure S3 caption: panel (a) is listed twice?

This will be corrected in the revised manuscript.

Reviewer #2 (Remarks to the Author):

The authors develop Nanopore based DMS-MaPseq to detect isoform specific RNA structure. The benefits of Nanopore sequencing are that it works on very long molecules and it can directly sequence RNA and in principle detect the DMS methylation without any conversion. The challenges are that nanopore sequencing is low accuracy relative to Illumina platforms, the base-callers require training, and it is expensive per read. The authors use some middle-ground where they sequence cDNA molecules. They troubleshoot conditions and find that Marathon +Manganese can read through the DMS modifications and produce long cDNA molecules. The authors do a lot of increase the signal over noise e.g. quality filtering, optimizing read depth, and taking only mismatches as signal. They demonstrate good agreement with known HIV structures and their data suggests structural implications on splicing. Overall, the work is novel and the approach is of vast interest. However, the authors must address some key questions before I would recommend publication:

1. How well does it work compared to highly accurate sequencing (e.g. Illumina platform)? It would be great if the authors can take the exact same cDNA library, fragment it and sequence it by established protocols to show that there is no/minimal bias and signal loss in their new sequencing platform.

This interesting experiment was also suggested by reviewer 1. We have already performed tagmentation and Illumina sequencing of the same cDNA library used for Nanopore sequencing, and are currently awaiting sequencing results. This will allow us to differentiate RT errors from Nanopore sequencing errors and to allow a more in-depth comparison between the methods.

2. The use of manganese is concerning because it makes the RT reaction error prone. The authors should compare an untreated control with and without manganese to assess how much noise comes from use of manganese.

The reverse transcription conditions were optimized by Guo et al. 2020 (Guo et al. 2020 Figure 3, PMID: 32259542). While error rates of untreated RNA indeed increased in presence of Mn, the signal to noise ratio between DMS-probed and unprobed RNA increased more strongly when using Mn compared to without. Therefore, despite the increase error rate, the advantage of including Mn outweighs the disadvantages.

3. It would be great if the authors can compare Marathon +Mn with the one that has been published previously using TGIRT enzyme (Zubrad et al), which avoids Manganese.

MarathonRT and TGIRT are both very processive intron group II reverse transcriptases. A comparison between these two enzymes is of interest, but extensive comparisons between MarathonRT and TGIRT have already been performed by the Pyle group (Zhao et al. 2018, PMID: 29109157) [albeit without DMS modification].

4. Since HIV-1 has been probed with DMS in virion and in cells (Tomezsko et al 2020) the authors should directly compare DMS signal over the 5'UTR and if available over the whole genome inside the virion (where there is just the full length molecule) with published results. In principle the two strategies should produce the exact same DMS signal and if there are discrepancies it would be important to know the source.

This is a good idea. As discussed above, we are including Nano-DMS-Map data for the entire HIV-1 genome in the discussion, and we will include the requested comparisons (Figure R3).

In a preliminary analysis we show a correlation of 0.82 between Nano-DMS-MaP and DMS-MaP reactivities extracted from Tomezsko *et al* of the RRE probed in virions (cluster 1). Moreover, Nano-DMS-MaP reactivities nearly perfectly agree with the consensus structure of the RRE.

5. Somethings that were not clear to me – what is the longest piece that was sequenced with high quality data, can the authors get the entire HIV-1 genome on one molecule? It seemed that the majority of the data is coming from 500-1kb, please explain. Also, a histogram of number of mutations per molecule would be very useful.

In our original submission, we obtained structural data for molecules 500-1.6kb, which was sufficient to perform isoform resolved structural analysis for our purposes. Nevertheless, we agree (also with reviewer 1) that an important advantage of our method is the ability to sequence long RNA molecules. We therefore now provide additional DMS data for the entire HIV-1 genome showing that Nano-DMS-MaP can analyze RNA molecules >4kb. These data will be provided in the revised manuscript (Figure R3).

Figure R3: Nano-DMS-MaP reactivity for the entire HIV-1 genome in virions (native and deproteinated). These data were obtained by Nano-DMS-MaP-seq of two amplicons of 4.5kb.

The modification density per molecule for all tested DMS concentrations will also be included in the revised manuscript.

Decision Letter, first revision:

Dear Redmond,

Thank you for submitting your revised manuscript "Nano-DMS-MaP-seq allows isoform specific RNA structure determination" (N METH-A49807B). It has now been seen by the original referees and their comments are below. The reviewers find that the paper has improved in revision, and therefore we'll be happy in principle to publish it in Nature Methods, pending minor revisions to satisfy the referees' final requests and to comply with our editorial and formatting guidelines.

We found your revision plan to be appropriate, and ask that you add the additional clarifications in response to the reviewers during your revision.

TRANSPARENT PEER REVIEW

Nature Methods offers a transparent peer review option for new original research manuscripts submitted from 17th February 2021. We encourage increased transparency in peer review by publishing the reviewer comments, author rebuttal letters and editorial decision letters if the authors agree. Such peer review material is made available as a supplementary peer review file. Please state in the cover letter 'I wish to participate in transparent peer review' if you want to opt in, or 'I do not wish to participate in transparent peer review' if you don't. Failure to state your preference will result in delays in accepting your manuscript for publication.

ORCID

IMPORTANT: Non-corresponding authors do not have to link their ORCIDs but are encouraged to do so. Please note that it will not be possible to add/modify ORCIDs at proof. Thus, please let your co-authors know that if they wish to have their ORCID added to the paper they must follow the procedure

described in the following link prior to acceptance:

Sincerely,

Rita

Rita Strack, Ph.D.

Senior Editor

Nature Methods

Reviewer #1 (Remarks to the Author):

The authors have addressed many of the prior criticisms and the manuscript is significantly improved. The new data collected on a broader panel of RNAs is impressive, and clearly establishes the generality of Nano-DMS-MaP. The data collected on 4 kb amplicons from the full-length HIV genomic RNA is also impressive. With that said, I still have several remaining concerns:

-The authors argue that quantifying performance of DMS-guided structure predictions is outside the scope of this work. I agree that modeling algorithms are imperfect, but existing SHAPE and DMS strategies are still able to give reasonably accurate answers, and the authors rely on such modeling strategies in their paper. It is thus reasonable to quantify whether Nano-DMS-MaP data performs comparably for this task as other methods. The authors already have the needed data. Computing standard metrics such as base-pairing sensitivity and positive predictive value (PPV) should be trivial.

-The authors present exciting new data for the full-length genomic RNA data, but this data is minimally analyzed besides establishing that they are able to recapitulate a limited number of well-known motifs. This seems like a missed opportunity. At least a cursory analysis that attempts to draw general insights about the global structure of the RNA, or comparison with prior DMS/SHAPE experiments would benefit the field.

-The conclusion that Illumina generates more mismatches than nanopore sequencing in Supporting Figure 5 is surprising. The excess mismatch error rates in Illumina suggest a sequencing error rate of

~0.003 (at 0 DMS condition). Assuming the Illumina data is Q>30, this sequencing error rate is unlikely. The authors should provide a deeper discussion/explanation of this result.

-In Supporting Figure 15, the authors note that nanopore sequencing yields substantially more mutations per read. However, this analysis seems incongruent with the data shown in Supporting Figure 5 -- it does not make mathematical sense that Illumina sequencing would show clear DMS concentration dependence in Fig S5 and higher mutation rates than nanopore, but then show fewer modifications per molecule.

-The 2nd-to-last sentence that translation could be the cause of SL1 unfolding is a key caveat that I missed before. Ideally, the authors would have tested this via other experiments to actually establish that SL1 unfolding is deterministic of packaging. In the absence of further experiments, a clear statement such as "testing whether SL1 unfolding drives packaging is deterministic of RNA packaging fate is a key topic for future studies" should be added to the final sentence.

Minor comments

-The authors should consider adding supporting figure 7 panel A to main figure 2. The ROC-AUC of the HIV RNA and 18S RNA could also be added to fully summarize their validation.

-In Supporting Figure 9, there are panels a and c, but no b. The heatmap in the top right is also not labeled or explained in the caption.

-Supporting Figure 10 is challenging to read with the numbers upside down.

-Figure 4h – caption does not specify which sample this is (DMS treated?)

Reviewer #2 (Remarks to the Author):

The authors made substantial revisions that have improved the manuscript- comparing to Illumina and including more positive control RNAs as well as longer sequencing. My only remaining question in Supp Fig 6D, why is the correlation (between Illumina and nanopore) at A and C much higher than the correlation at G and Ts? Also, it is more intuitive if r^2 is showed instead of R.

Reviewer #3 (Remarks to the Author):

In this revision, Bohn and co-workers make a variety of impressive advances. First, they now benchmark on 5 structured RNAs with well known structures, as well as the 18S rRNA. They show correlation of

DMS data and Nanopore data Supp Fig 6d, attained by excluding indel data and including only high Q-score from the Nanopore reads. They perform analysis on 16 spliced viral transcripts of HIV-1 and determine the structure of the HIV-genome in its entirety in cells and virions. The authors also addressed my other concerns including. Overall, this is an excellent study that I now support for publication in Nature Methods.

Author Rebuttal, first revision:

Reviewer #1 (Remarks to the Author):

The authors have addressed many of the prior criticisms and the manuscript is significantly improved. The new data collected on a broader panel of RNAs is impressive, and clearly establishes the generality of Nano-DMS-MaP. The data collected on 4 kb amplicons from the full-length HIV genomic RNA is also impressive. With that said, I still have several remaining concerns:

- The authors argue that quantifying performance of DMS-guided structure predictions is outside the scope of this work. I agree that modeling algorithms are imperfect, but existing SHAPE and DMS strategies are still able to give reasonably accurate answers, and the authors rely on such modeling strategies in their paper. It is thus reasonable to quantify whether Nano-DMS-MaP data performs comparably for this task as other methods. The authors already have the needed data. Computing standard metrics such as base-pairing sensitivity and positive predictive value (PPV) should be trivial.

We now provide base-pairing sensitivity and positive predictive value metrics for the 5'UTR genomic RNA model in extended data figure 5f and 10g.

Please note that this is the only structure obtained by *de novo* folding where the 'ground truth' is available, and thus the only structure for which we could perform this analysis.

-The authors present exciting new data for the full-length genomic RNA data, but this data is minimally analyzed besides establishing that they are able to recapitulate a limited number of well-known motifs. This seems like a missed opportunity. At least a cursory analysis that attempts to draw general insights about the global structure of the RNA, or comparison with prior DMS/SHAPE experiments would benefit the field.

We thank the reviewer for their kind words.

Our current analysis already provides four major conclusions regarding the *local* and *global* folding of the RNA:

1. Key local RNA structures, such as the 5'UTR, frameshifting site and RRE are present in both cells and virions (Fig. 3b-d).
2. Comparison of cellular and virion RNA DMS reactivities can identify local changes due to intermolecular interactions at the PBS and SL1 (Sup. Fig. 2).
3. DMS reactivities are globally increased in the virion compared to the cells, suggesting the the RNA is more relaxed in the virion, which we speculate may facilitate reverse transcription (Fig. 3a).
4. The correlation between replicates in the cell is lower than that of the virus, suggesting that global viral RNA structure is more heterogeneous in cells (Extended Data Fig. 8a-b).

Points 1 and 2 demonstrate that the data is robust, by recapitulating known information about HIV-1 RNA structure. Points 3 and 4 provide novel insights that can be tested in future virological assays.

As mentioned in a previous response, there are no publically available DMS data sets that would allow a direct comparison of Nano-DMS-MaP reactivities with those obtained with an another technique e.g. DMS-MaPseq. For the reviewer's interest, however, we performed cursory

global reactivity correlations between DMS and several SHAPE data sets, with caveats that the HIV constructs differ in parts of the env ORF sequence, that probing has been performed in different environments (for example RNA extraction and refolding), and that different probing reagents have different reaction kinetics towards different nucleotides.

Even with these caveats, we find positive Spearman's correlation within not only datasets of the same reagent, but also between the DMS and SHAPE datasets, suggesting that both measure a similar underlying structure.

Figure 1: Correlation between reactivity profiles of A and C residues between our DMS probing data and published datasets.

We feel that leaning further into the data would require extensive validation experiments that are beyond the scope of the current study, which is why we are provided the measured reactivities as a supplementary data set for use in RNA structure prediction algorithms by the global community.

The conclusion that Illumina generates more mismatches than nanopore sequencing in Supporting Figure 5 is surprising. The excess mismatch error rates in Illumina suggest a sequencing error rate of ~0.003 (at 0 DMS condition). Assuming the Illumina data is Q>30, this sequencing error rate is unlikely. The authors should provide a deeper discussion/explanation of this result.

This observation can be explained by three factors, namely other sources of mismatch errors, median vs mean expected error rate, and aligner choice:

1. A score of Q30 is the expected sequencing error rate (0.1%).

This metric does not take into account errors introduced during biological processes (e.g. potential RNA modifications) and those introduced during library preparation, such as reverse transcription with Mn^{2+} (known to be error prone) and 25 cycles of PCR. Thus, the observed error rate can be higher than predicted by the Q-score in such experimental setups.

- Whilst our NovaSeq Illumina data set has a *median* Q-score of Q37 across all positions and per read, signifying a high-quality sequencing run, the *mean* Q-score is Q27 when applying identical filtering steps as those used for Nanopore data (*mean read Phred Q-score above 10 and per position Qscore above 22*). Thus, high quality Illumina data can still contain some lower quality nucleotides that are not represented in the widely used *median* Q-score metric, and which ultimately result in a *mean* expected mismatch rate of 0.18% (Figure 2).

Figure 2: (a) Median and Mean Phred Qscore per position in read of the Illumina NovaSeq sequencing run. (b) Median Phred Qscore per read distribution. (c) Mean Phred Qscore per read distribution. Mean Phred Qscores were calculated by converting Qscores per position (a) or per read (c) to their error probability, calculating the mean error rate, then converting back to Phred scores.

- Another factor in determining the shown mismatch rates (and mutational profiling analysis) is the alignment. While a more stringent alignment reduces measured mismatch rates, it may also negatively affect mutational profiling signal. We have evaluated both LAST and bowtie for alignment of the Illumina data, with LAST resulting in more stringent alignments (<0.1% mismatch rate, even for non-filtered data), showing that aligner choice has a major impact on mismatch rate evaluation. Critically, the more stringent alignment however also resulted in decreased correlation between replicates, and lower ROC-AUC scores, suggesting that bowtie is the better choice here, and that MaP signal can be negatively affected by too stringent alignment.
- To evaluate the impact of lower quality reads, we also performed pre-filtering for reads with mean Qscore above 30, however this did not notably affect ROC-AUC scores, and observed mismatch rates (aligned with bowtie) were also nearly equal.

In conclusion, the *mean* quality scores (Q27) in our experiment match well with the observed mean error rate (0.3%).

-In Supporting Figure 15, the authors note that nanopore sequencing yields substantially more mutations per read. However, this analysis seems incongruent with the data shown in Supporting Figure 5 -- it does not make mathematical sense that Illumina sequencing would

show clear DMS concentration dependence in Fig S5 and higher mutation rates than nanopore, but then show fewer modifications per molecule.

Although it might seem paradoxical, this result is explained by the longer read length of nanopore sequencing compared to Illumina. We updated Sup. Fig. 5 to also show the read length distributions (rightmost panel).

While the mean aligned read length (subtracted by primer sequences) of the unspliced 85 mM Nanopore reads is 444 nt, the mean length of Illumina reads is only 129 nt.

Thus, while the mean mutation rate in Illumina data is higher at 1.75% compared to 1.07% for Nanopore data, the mean number of mutations per read in the Illumina dataset comes out at 2.2, while it is 4.7 for Nanopore reads.

-The 2nd-to-last sentence that translation could be the cause of SL1 unfolding is a key caveat that I missed before. Ideally, the authors would have tested this via other experiments to actually establish that SL1 unfolding is deterministic of packaging. In the absence of further experiments, a clear statement such as "testing whether SL1 unfolding drives packaging is deterministic of RNA packaging fate is a key topic for future studies" should be added to the final sentence.

We have edited the final sentence to include this hypothesis.

Minor comments

-The authors should consider adding supporting figure 7 panel A to main figure 2. The ROC-AUC of the HIV RNA and 18S RNA could also be added to fully summarize their validation.

This is a good suggestion. We have moved the panel A containing the ROC-AUCs to the main Figure 2.

-In Supporting Figure 9, there are panels a and c, but no b. The heatmap in the top right is also not labeled or explained in the caption.

We have added the missing label and description.

-Supporting Figure 10 is challenging to read with the numbers upside down.

We have rotated the text.

-Figure 4h – caption does not specify which sample this is (DMS treated?)

We added details to clarify that this is shown for samples across all DMS concentrations.

Reviewer #2 (Remarks to the Author):

The authors made substantial revisions that have improved the manuscript- comparing to Illumina and including more positive control RNAs as well as longer sequencing. My only remaining question in Supp Fig 6D, why is the correlation (between Illumina and nanopore) at A and C much higher than the correlation at G and Ts?

Supp. Fig. 6D (now Extended Data Fig 5d) shows the correlation of reactivities between nanopore and Illumina DMS MaP data sets.

G and U residues have lower mutation rates than A and C residues, and are therefore subject to higher noise when calculating DMS reactivities. This higher noise therefore explains the lower correlation.

Also, it is more intuitive if r^2 is showed instead of R.

We have updated the figures to show the coefficient of determination (r^2) instead of the pearson correlation coefficient (R)

Reviewer #3 (Remarks to the Author):

In this revision, Bohn and co-workers make a variety of impressive advances. First, they now benchmark on 5 structured RNAs with well known structures, as well as the 18S rRNA. They show correlation of DMS data and Nanopore data Supp Fig 6d, attained by excluding indel data and including only high Q-score from the Nanopore reads. They perform analysis on 16 spliced viral transcripts of HIV-1 and determine the structure of the HIV-genome in its entirety in cells and virions. The authors also addressed my other concerns including. Overall, this is an excellent study that I now support for publication in Nature Methods.

Signed: Philip Bevilacqua

This email has been sent through the Springer Nature Tracking System NY-610A-NPG&MTS

Final Decision Letter:

Dear Redmond,

I am pleased to inform you that your Article, "Nano-DMS-MaP-seq allows isoform specific RNA structure determination", has now been accepted for publication in Nature Methods. Your paper is tentatively scheduled for publication in our May print issue, and will be published online prior to that. The received and accepted dates will be July 12, 2022 and March 21, 2023. This note is intended to let you know what to expect from us over the next month or so, and to let you know where to address any further questions.

Once your paper is typeset, you will receive an email with a link to choose the appropriate publishing options for your paper and our Author Services team will be in touch regarding any additional information that may be required.

Please note that *Nature Methods* is a Transformative Journal (TJ). Authors may publish their research with us through the traditional subscription access route or make their paper immediately open access through payment of an article-processing charge (APC). Authors will not be required to make a final decision about access to their article until it has been accepted. [Find out more about Transformative Journals](https://www.springernature.com/gp/open-research/transformative-journals)

Your paper will now be copyedited to ensure that it conforms to Nature Methods style. Once proofs are generated, they will be sent to you electronically and you will be asked to send a corrected version within 24 hours. It is extremely important that you let us know now whether you will be difficult to contact over the next month. If this is the case, we ask that you send us the contact information (email, phone and fax) of someone who will be able to check the proofs and deal with any last-minute problems.

If, when you receive your proof, you cannot meet the deadline, please inform us at rjsproduction@springernature.com immediately.

Once your manuscript is typeset and you have completed the appropriate grant of rights, you will receive a link to your electronic proof via email with a request to make any corrections within 48 hours. If, when you receive your proof, you cannot meet this deadline, please inform us at rjsproduction@springernature.com immediately.

Once your paper has been scheduled for online publication, the Nature press office will be in touch to confirm the details.

Once your paper has been scheduled for online publication, the Nature press office will be in touch to confirm the details.

Content is published online weekly on Mondays and Thursdays, and the embargo is set at 16:00 London time (GMT)/11:00 am US Eastern time (EST) on the day of publication. If you need to know the exact publication date or when the news embargo will be lifted, please contact our press office after you have submitted your proof corrections. Now is the time to inform your Public Relations or Press Office about your paper, as they might be interested in promoting its publication. This will allow them time to prepare an accurate and satisfactory press release. Include your manuscript tracking number NMETH-A49807C and the name of the journal, which they will need when they contact our office.

About one week before your paper is published online, we shall be distributing a press release to news organizations worldwide, which may include details of your work. We are happy for your institution or

funding agency to prepare its own press release, but it must mention the embargo date and Nature Methods. Our Press Office will contact you closer to the time of publication, but if you or your Press Office have any inquiries in the meantime, please contact press@nature.com.

Nature Portfolio journals [encourage authors to share their step-by-step experimental protocols](https://www.nature.com/nature-research/editorial-policies/reporting-standards#protocols) on a protocol sharing platform of their choice. Nature Portfolio 's Protocol Exchange is a free-to-use and open resource for protocols; protocols deposited in Protocol Exchange are citable and can be linked from the published article. More details can found at www.nature.com/protocolexchange/about.

Please note that you and any of your coauthors will be able to order reprints and single copies of the issue containing your article through Nature Portfolio 's reprint website, which is located at <http://www.nature.com/reprints/author-reprints.html>. If there are any questions about reprints please send an email to author-reprints@nature.com and someone will assist you.

Please feel free to contact me if you have questions about any of these points. It's been a pleasure working with you on this paper.

Best regards,
Rita

Rita Strack, Ph.D.
Senior Editor
Nature Methods